# Roles of P-body factors in *Candida albicans* filamentation and stress response

**Melissa A. Tosiano[1], Frederick Lanni[1], Aaron P. Mitchell[2], C. Joel McManus[1]***

**1** Department of Biological Sciences, Carnegie Mellon University, Pittsburgh, Pennsylvania, United States of America, **2** Department of Microbiology, University of Georgia, Athens, Georgia, United States of America

* mcmanus@andrew.cmu.edu

**Data availability statement:** Raw RNA-seq data have been deposited at the NCBI SRA

## Abstract

Hyphal growth is strongly associated with virulence in the human fungal pathogen *Candida albicans*. While hyphal transcriptional networks have been the subject of intense study, relatively little is known about post-transcriptional regulation. Previous work reported that P-Body (PB) factors Dhh1 and Edc3 were required for *C. albicans* virulence and filamentation, suggesting an essential role for post-transcriptional regulation of these processes. However, the molecular roles of these factors have not been determined. To further study the function of PB factors in filamentation, we generated homozygous deletions of *DHH1* and *EDC3* in diverse prototrophic clinical strains using transient CRISPR-Cas9. Homozygous *DHH1* deletion strongly impaired growth, altered filamentation, and exhibited unusual colony morphology in response to heat stress in five strain backgrounds. Using RNA-seq, we found *DHH1* deletion disrupts the regulation of thousands of genes under both yeast and hyphal growth conditions in SC5314 and P57055. This included upregulation of many stress response genes in the absence of external stress, similar to deletion of the *S. cerevisiae DHH1* homolog. In contrast, we found *EDC3* was not required for heat tolerance or filamentation in diverse strains. These results support a model in which *DHH1*, but not *EDC3*, represses hyphal stress response transcripts in yeast and remodels the transcriptome during filamentation. Our work supports distinct requirements for specific mRNA decay factors, bolstering evidence for post-transcriptional regulation of filamentation in *C. albicans*.

## Author summary

In the dimorphic fungal pathogen *C. albicans*, the hyphal phenotype strongly correlates with pathogenicity. While transcriptional control of hyphal growth has been extensively studied, comparatively little is known about post-transcriptional regulation of this significant morphological shift. PB factors are associated with mRNA decay and translational repression. Here we investigate the roles of two PB factors in growth, filamentation, and gene expression. Although deletion of PB factor *EDC3* did not impact growth or filamentation, *dhh1Δ/Δ* had greatly impaired growth and heat tolerance as well as unusual hyphal phenotypes in multiple clinical isolate strains. Additionally, we found that the

under bioproject number PRJNA1133750. Processed expression data are available at NCBI GEO under accession GSE284543. All other data are provided in the supplemental tables.

**Funding:** This study was supported by funding from the National Institutes of Health, USA, grants R35GM145317 and R21AI126222 to CJM and T32GM133353 to MAT. The funders had no role in study design, data collection and analysis, decision to publish, or preparation of the manuscript.

**Competing interests:** The authors have declared that no competing interests exist.

transcriptomes of SC5314 and P57055 *dhh1Δ/Δ* yeast and hyphae were highly dysregulated. The extensive transcriptomic impacts of the absence of Dhh1 correlated with our phenotypic findings. Stress-associated genes were induced under non-stress conditions and the filamentation response was blunted under physiologically relevant *in vitro* conditions. We demonstrate that mRNA decay factors play distinct roles in regulating *C. albicans* morphology and that Dhh1 contributes to environmentally appropriate expression of the stress response and hyphal growth.

## Introduction

*Candida albicans* is an opportunistic pathogen that causes blood and deep tissue infections (candidiasis) in immunocompromised individuals [1,2]. Pathogenicity requires responding to stress in divergent physiological microenvironments [3,4]. Under stress, *C. albicans* undergoes filamentation, shifting from commensal yeast to virulent hyphae. Hyphal growth correlates with the release of the peptide toxin candidalysin [5] and adherent biofilm formation [6–8]. Both contribute to pathogenicity. Foundational research on transcriptional regulation of filamentation identified dozens of transcription factors (TFs) and signaling cascades controlling core filamentation response networks [9–12]. Under different hyphal-inducing conditions, *C. albicans* activates distinct transcriptional programs [13]. Notably, these transcriptional programs have been found to vary dramatically across different clinical isolates, even under the same filamentation conditions [14]. Despite deep exploration into transcriptional control of filamentation, comparatively little work has addressed post-transcriptional regulation of filamentation [15,16].

Eukaryotes implement swift and precise post-transcriptional stress responses [17,18] appropriate for regulating the acute stage of *C. albicans* filamentation [15,16]. Post-transcriptional regulation of gene expression integrates mRNA translation, localization, and decay [19,20]. Indeed, mRNA decay has been found to regulate adaptation to the host environment and virulence in several human fungal pathogens [3,16,21,22]. Localization to ribonucleoprotein (RNP) granules represses translation and initiates decay in many organisms [23,24]. P-bodies (PBs) are RNPs associated with the Ccr4-not deadenylase complex [24–27] and mRNA degradation [23,24,28]. In *C. albicans*, several PB mRNA decay factors have been reported to function in hyphal development and virulence, including Ccr4, Pop2, Not5, Xrn1, Dhh1, and Edc3 [29–37]. However, all of these studies used auxotrophic derivatives of SC5314 such as BWP17 and CAI4, therefore it is unknown how important these factors are in clinical isolates. In addition, the effects of these factors on gene regulation during filamentation have not been elucidated.

Stochastic transcriptional noise generates different phenotypic outcomes in clonal microbial populations that serves as a mechanism of "bet-hedging" for acute environmental stressors [38–40]. In this context, mRNA decay provides a system of transcript buffering preventing gene expression under inappropriate conditions [41]. Genes identified as both transcriptionally noisy and poorly translated have these features conserved between *S. cerevisiae* and *C. albicans* homologs [42]. However, the extent to which mRNA decay factors buffer or control noisy expression of hyphal genes under yeast-form conditions or conversely yeast genes under hyphal conditions remains unclear in *C. albicans*. Furthermore, the roles of *C. albicans* PB factors in buffering noisy transcription for environmentally appropriate gene expression remain unknown.

In the model yeast *S. cerevisiae*, the functions of PB factors Dhh1 and Edc3 have been extensively studied [43]. The DEAD-box helicase and activator of mRNA decapping Dhh1 rapidly exchange between the cytoplasm and PBs [44,45], and may repress translation and

stimulate the decay of mRNAs with poor codon optimality [46,47]. Under optimal growth conditions it has been reported that Dhh1 represses genes associated with nutrient limitation, cell wall components, autophagy, and metabolism of non-preferred carbon and nitrogen sources [48]. Loss of *DHH1* in *S. cerevisiae* downregulates *STE12* translation leading to defects in pseudohyphal growth and mating [35,49,50], and the *C. neoformans* Dhh1 homolog (Vad1) plays a similar role in regulating mating [51]. Kinase signaling via Dhh1 phosphorylation sites are critical to pseudohyphal development in *S. cerevisiae* [35,52]. Edc3 is a non-essential enhancer of mRNA decapping and a PB scaffolding factor in *S. cerevisiae* [43,44,53,54]. In *S. cerevisiae*, the absence of Edc3 reduces PB assembly size, but does not impede global mRNA decay [44,55]. Additionally, Edc3 is dispensable for PB formation in *S. pombe* [56]. In contrast to *S. cerevisiae*, the roles of Dhh1 and Edc3 in *C. albicans* hyphal growth have been minimally explored and the impact of Dhh1 on gene expression remains unknown.

Prior work in *C. albicans* reported that both Dhh1 and Edc3 are necessary for WT stress tolerance and filamentation. In a *C. albicans* auxotrophic background, the authors described a heterozygous deletion of *DHH1* (*DHH1/dhh1Δ*) which produced WT colonies on YPD plates, but was filamentation impaired [35]. In the same *C. albicans* auxotrophic background, *edc3Δ/Δ* was noted to have compromised PB condensation, reduced heat tolerance at 37°C, and severely restricted filamentation [36,37]. This early research implied that *DHH1* and *EDC3* may have distinct roles in *C. albicans* filamentation. However, the gene regulatory effects of *DHH1* deletion in *C. albicans* have not been defined. Thus, our understanding of these important PB factors in filamentation remains limited.

In this study, we investigated the roles of PB factor Dhh1 in *C. albicans* heat tolerance, filamentation, and gene expression. Using transient CRISPR-cas9 [57] we deleted *DHH1* and *EDC3* in diverse clinical isolate strains representing the five major *C. albicans* clades [58]. We demonstrated the dispensability of *EDC3* for PB condensation, heat tolerance, and filamentation. Importantly, we found *dhh1Δ/Δ* exhibits impaired yeast-form growth, heat sensitivity, and anomalous filamentation across genetic backgrounds. Finally, we determined the effects of *dhh1Δ/Δ* on genome-wide expression in yeast and early hyphae, and found loss of *DHH1* causes widespread dysregulation of the transcriptome and stress responses in SC5314 and P57055.

## Results

### Characterizing P-Bodies (PBs) in *C. albicans*

To investigate the presence of Dhh1 and Edc3 in PBs, we first fused mScarlet to the C-terminus of Dhh1 (Dhh1-mS) and mNeonGreen to the C-terminus of Edc3 (Edc3-mNG) using CRISPR-Cas mediated genome editing in strain SC5314. Under optimal temperature and acute heat shock, Edc3-mNG condensed into round foci (Fig 1A–B). To determine if these foci were PBs, we generated a dual labeled strain (Edc3-mNG, Dhh1-mS). Edc3-mNG and Dhh1-mS colocalized, confirming these foci were PBs (Figs 1C and S1A). As is the case in *S. cerevisiae* [59], we found a large fraction of Dhh1 was cytoplasmically diffuse, suggesting it functions both in PBs and in the cytoplasm (Figs 1C and S1A). We next investigated PB condensation during early filamentation. After 3 hours of acute starvation in PBS, PBs were abundant in yeast cells and along developing hyphal germ tubes (Fig 1D). PBs also formed and persisted over 4 hours of filamentation (37°C and RPMI), (Fig 1E). Lastly, we investigated whether these proteins were independently required for PB condensation. In the absence of Edc3, Dhh1-mS retained the capacity to form condensates in response to acute heat shock (Figs 1F and S1B). Similarly, Edc3-mNG also condensed in the absence of Dhh1 (Figs 1G and S1C). In summary, we confirmed the presence of PBs in *C. albicans* and found that, as in

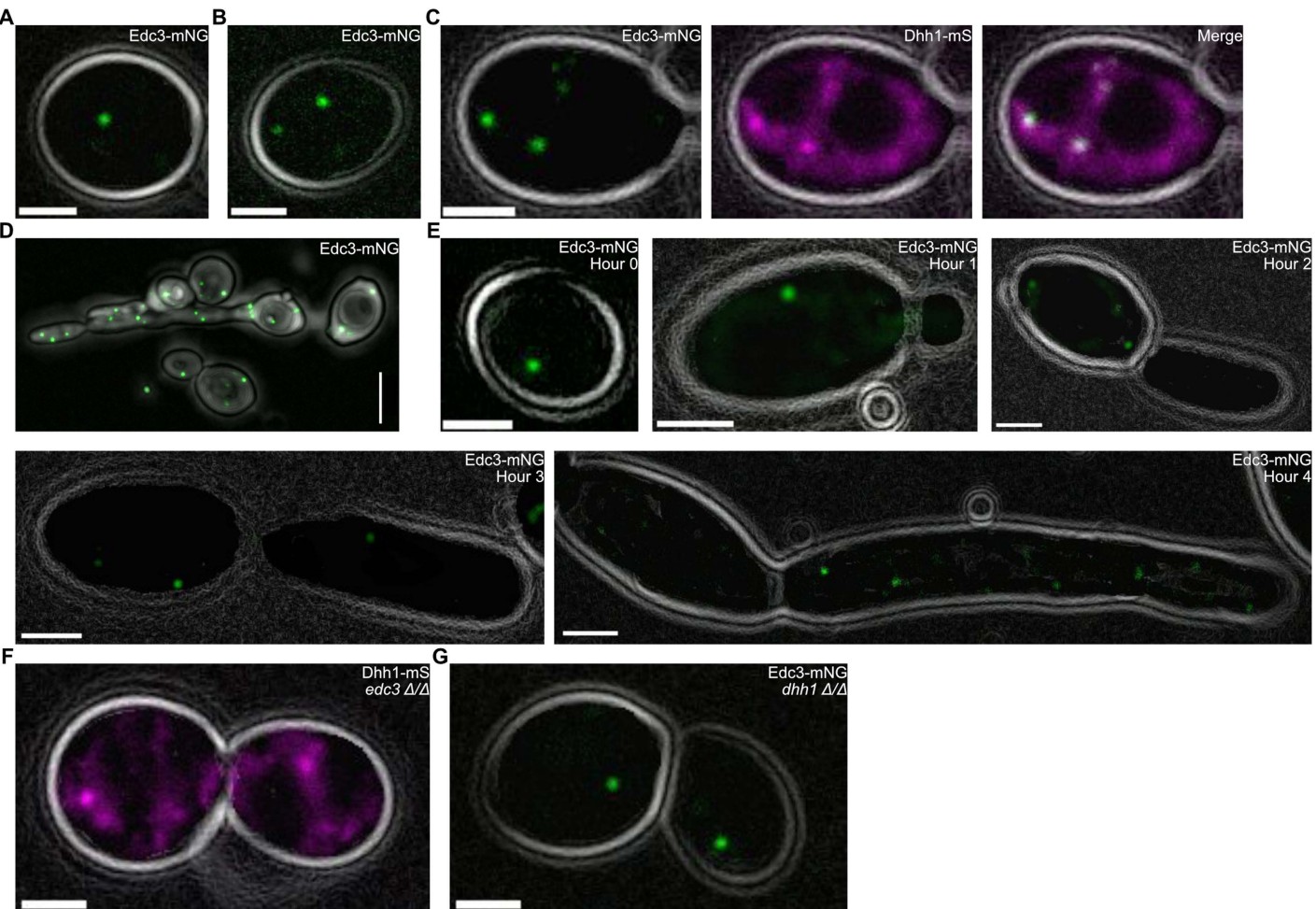

**Fig 1. PBs condense during acute stress.** Images of PB factors fluorescently tagged in strain SC5314. A. Edc3-mNG condensed in live *C. albicans* under non-stress conditions (30°C in complete media (CM)). Scale bar = 5μm. B. Edc3-mNG condensed in live *C. albicans* in response to acute heat shock (10 minutes at 46°C in CM). Scale bar = 5μm. C. PB factors (Edc3-mNG and DHH1-mS) co-localized during acute heat shock (10 minutes at 46°C in CM) confirming their identity as PBs. Scale bar = 2μm. D. After 3-hours of starvation in PBS at room temperature, abundant PBs (Edc3-mNG) were visible with confocal microscopy along the developing germ tube. Scale bar = 5μm. E. PBs (Edc3-mNG) also condensed and persisted during hyphal growth (37°C in RPMI). Scale bars = 2μm. F. Dhh1-mS condensed in the absence of Edc3 in response to acute heat shock (10 minutes at 46°C in CM). Scale bar = 2μm. G. Dhh1 is not required for Edc3-mNG PB formation in response to acute heat shock (10 minutes at 46°C in CM). Scale bar = 5μm.

*S. cerevisiae*, *C. albicans* PBs are robust condensates tolerating the absence of Dhh1 or Edc3 individually.

### *DHH1* is required for normal growth and stress response phenotypes in SC5314

We next investigated the role of Dhh1 and Edc3 in *C. albicans* growth and stress responses. We generated heterozygous (*DHH1/dhh1Δ*) and homozygous (*dhh1Δ/Δ*) *DHH1* deletions and complements in SC5314. Initially, we conducted spot assays at 30°C and 37°C to test for growth and temperature sensitivity defects. Strains possessing at least one copy of *DHH1* produced colonies identical to WT on YPD plates (Fig 2A-B). However, *dhh1Δ/Δ* had impaired growth across timepoints and incubation temperatures (Fig 2A-B). We also observed slow growth of *dhh1Δ/Δ* at 30°C in liquid YPD over 12 hours (Fig 2C). Uniquely, after 3 days on

YPD at 37°C, *dhh1Δ/Δ* colonies exhibited a wrinkled appearance associated with hyphal growth (Fig 2B). To further investigate, we grew colonies at 37°C for 4 days on YPD. While strains with *DHH1* had a smooth surface and soft texture, *dhh1Δ/Δ* colonies had a wrinkled surface and firm, dry texture reminiscent of hyphal colonies on spider media (Fig 2D). Surprisingly, DIC microscopy revealed *dhh1Δ/Δ* cells were twice the size of the *DHH1*-containing strains, with enlarged vacuoles (Fig 2E). The size of the cells (Fig 2F), vacuoles (Fig 2G), and proportion of the cell taken up by the vacuole (Fig 2H) were all significantly greater in *dhh1Δ/Δ* cells. The slow growth and altered cell physiology of *dhh1Δ/Δ* suggested the induction of stress associated phenotypes in the absence of environmental stressors. In contrast, the loss of *EDC3* did not impair colony growth at 30°C and 37°C in SC5314 (S2A Fig). Thus, we focused the remainder of this study on *DHH1*.

### *DHH1* is necessary for SC5314 WT filamentation

We next examined filamentation in *dhh1Δ/Δ*. SC5314 *dhh1Δ/Δ* cells were larger than WT and complement strains after 4 hours of filamentation, with the cell body taking up a greater proportion of the total length (Fig 3A). However, the median length to width ratio of *dhh1Δ/Δ* cells was significantly smaller, due to impaired germ tube elongation (Fig 3B). After 16 hours of planktonic filamentation, *dhh1Δ/Δ* cells remained enlarged with thick curved germ tubes and fewer septa compared to WT and complement strains (Fig 3C). The median length to width ratio of *dhh1Δ/Δ* at 16 hours was also significantly less than the other strains (Fig 3D). In summary, we found that a single copy of *DHH1* sufficiently supported WT planktonic filamentation, while homozygous deletion resulted in abnormal planktonic filamentous growth.

Due to the significant impairment of planktonic filamentation in SC5314 *dhh1Δ/Δ,* we expected *dhh1Δ/Δ* to exhibit comparatively short defective biofilms. Surprisingly, *dhh1Δ/Δ* generated deep, diffuse biofilms compared to WT and complement strains (Fig 4A-B). Nevertheless, *dhh1Δ/Δ* biofilms exhibited other key differences. In WT and *DHH1* complement strains, cells were concentrated near the base of the biofilm (Fig 4C). Using average pixel intensity as a proxy for cell density, we quantified biofilm architecture, and found *dhh1Δ/Δ* biofilms were altered with a maximum cell density further removed from the basal layer of the biofilm compared to WT and *DHH1* complement (Fig 4D). This modified biofilm structure may result from the somewhat hyperfilamentous *dhh1Δ/Δ* strain, via decreased yeast cell proliferation at the base of the structure.

Next, we surveyed the phenotype of colonies during invasive growth on spider media. After 6 days of growth at 37°C, *dhh1Δ/Δ* colonies revealed a wrinkled hyphal phenotype but a marked absence of hyperfilamentous margins found in WT SC5314 (Fig 4E). Compared to strains with at least one copy of *DHH1*, *dhh1Δ/Δ* colonies grew more vertically with tighter wrinkles (Fig 4E). Again as in the other filamentation assays, our complement strain with a single *DHH1* copy produced phenotypically WT colonies on spider media (Fig 4E). Furthermore, in contrast to *dhh1Δ/Δ, edc3Δ/Δ* was not filamentation impaired in all three filamentation assays (S2B-F Fig). In conclusion, SC5314 *dhh1Δ/Δ* failed to generate WT germ tubes, biofilm composition or invasive colonies on spider media supporting a role for this RNA decay factor in filamentation.

### Loss of DHH1 disrupts numerous pathways involved in metabolism and cellular morphology in SC5314

Given the severe phenotypic defects in SC5314 *dhh1Δ/Δ* yeast and hyphae, we next used RNA-seq to identify differentially expressed genes in yeast (30°C, YPD) and early hyphae (37°C,

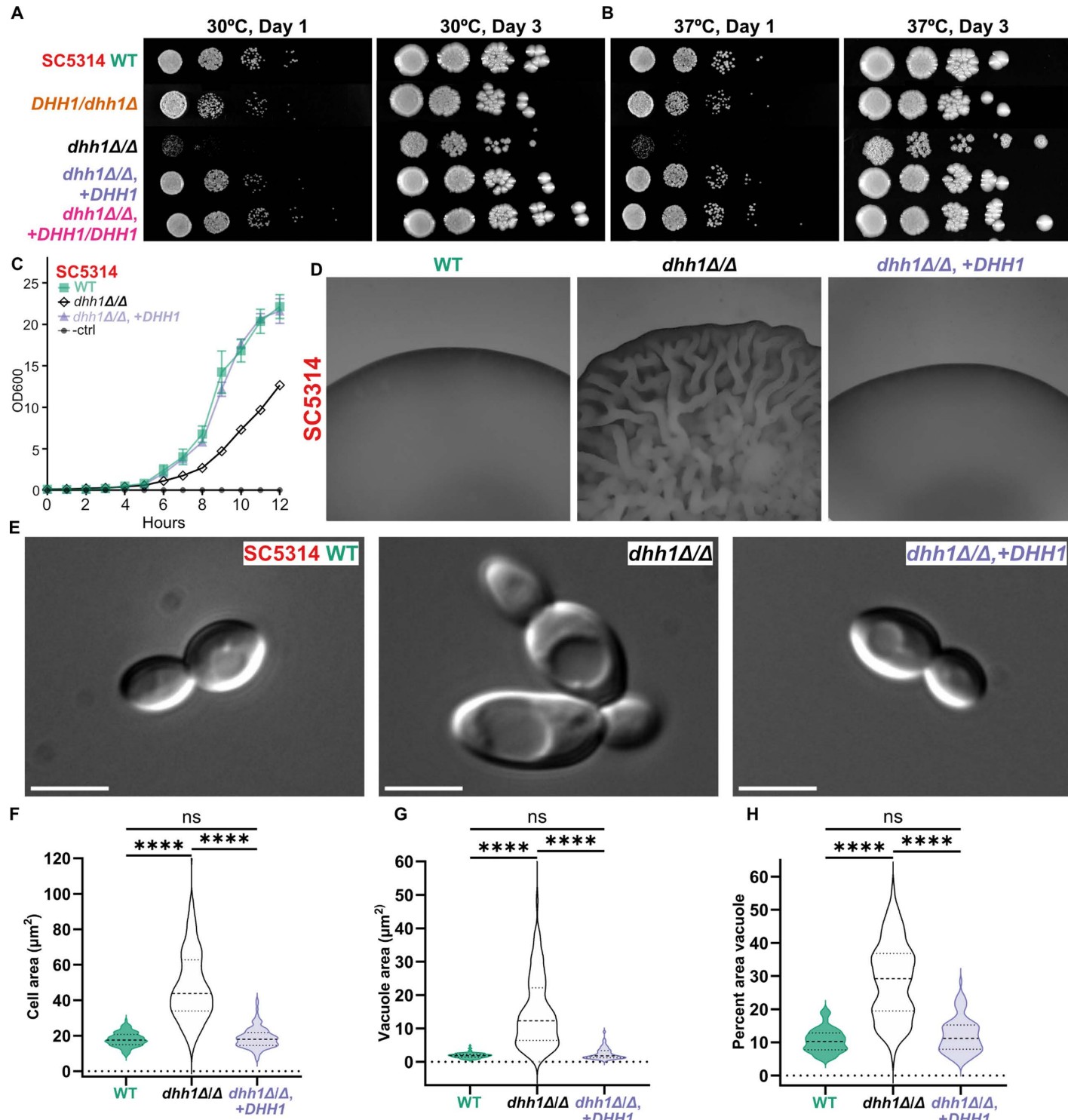

**Fig 2. DHH1 is required for WT growth and morphology in SC5314.** A. and B. *dhh1Δ/Δ* resulted in slow growth at 30°C and 37°C on YPD plates, which was complemented by a single copy of *DHH1*. C. *dhh1Δ/Δ* had reduced growth in liquid YPD (30°C). WT and mutant strains were grown in triplicate. Impaired *dhh1Δ/Δ* exponential growth was rescued by complementation (*dhh1Δ/Δ, +DHH1*). Error bars show 95% CI. D. Colonies lacking *DHH1* have a wrinkled morphology (4 days, YPD agar, 37°C). 25X magnification. E. DIC images of exponentially growing cells (YPD, 30°C) show enlarged cell bodies and vacuoles in *dhh1Δ/Δ*. Scale bars = 5μm. F.- H. Quantification of cell and vacuolar enlargement in *dhh1Δ/Δ* (N = 100 cells per strain; ****P < 0.0001, Kruskal-Wallis tests).

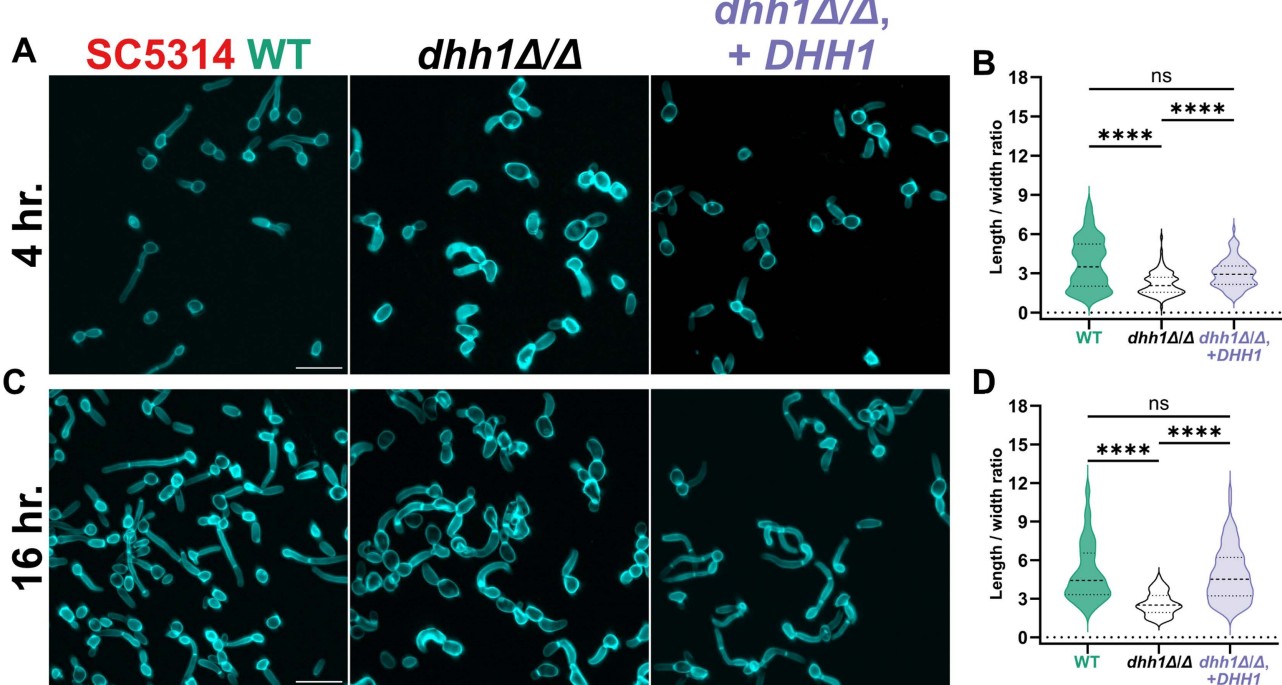

**Fig 3. DHH1 is required for wildtype planktonic filamentation.** WT and mutant strains grown under physiologically relevant filamentation conditions (37°C, RPMI + 10% FBS). A. and B. After 4-hours of filamentation, *dhh1Δ/Δ* had reduced germ tube development defined by the median ratio of cell length to width. C. and D. After 16-hours of filamentation, *dhh1Δ/Δ* had less developed germ tubes with fewer septa compared to wildtype and complement strains. (Scale bars = 20μm; N = 100 cells, ****P < 0.0001, Kruskal-Wallis tests).

RPMI + 10% FBS, 4 hours) in triplicate (see methods for details) [60]. After removing poorly transcribed genes (S8A and S8B Fig) we evaluated samples by PCA (S9A Fig) and differential expression using DEseq2 [61]. In YPD, *dhh1Δ/Δ* yeast had 1,244 genes significantly downregulated and 1,408 genes significantly upregulated (Padj < 0.01) compared to wildtype (Fig 5A and S1 Table). Interestingly, *dhh1Δ/Δ* yeast upregulated multiple genes associated with hyphae inducing stress, including *RHR2*, *ECE1*, and *HWP1* (Figs 5A, S3C, and S3D). Indeed, GO terms corresponding to hyphal stress, such as hyphal cell wall and carbohydrate transport, were enriched in upregulated genes (Fig 5B). In RPMI, *dhh1Δ/Δ* hyphae had 1,151 genes downregulated and 1,269 genes upregulated (Padj < 0.01) (Fig 5C and S2 Table). Lactate and sugar transport genes were upregulated in *dhh1Δ/Δ* RPMI including *JEN1* and *HGT12* (Fig 5C). Under these conditions, *dhh1Δ/Δ* inappropriately expressed many cell wall-associated transcripts. Hyphal adhesin *ALS1* was downregulated while yeast cell wall protein *YWP1* was upregulated (Figs 5C and S3E). Genes upregulated in *dhh1Δ/Δ* hyphae were enriched in GO terms related to transport, metabolism, and ergosterol biosynthesis (Fig 5D). While most changes in expression were specific to environmental conditions, there was significant overlap in the genes affected by *dhh1Δ/Δ* during growth in YPD and RPMI (Hypergeometric test, P < 2.3e$^{-22}$; Fig 5E). Transcription factors regulating cell morphology were also differentially expressed *dhh1Δ/Δ* in both conditions (Fig 5F). Interestingly, while most TFs shifted in the same direction in both conditions, hyphal repressor *NRG1* [62,63] and cell cycle/cell wall stress regulator *SWI6* [64,65] were downregulated in YPD but upregulated in RPMI (Fig 5F). In summary, RNA-seq established widespread gene expression aberrations in SC5314 *dhh1Δ/Δ* yeast and hyphae.

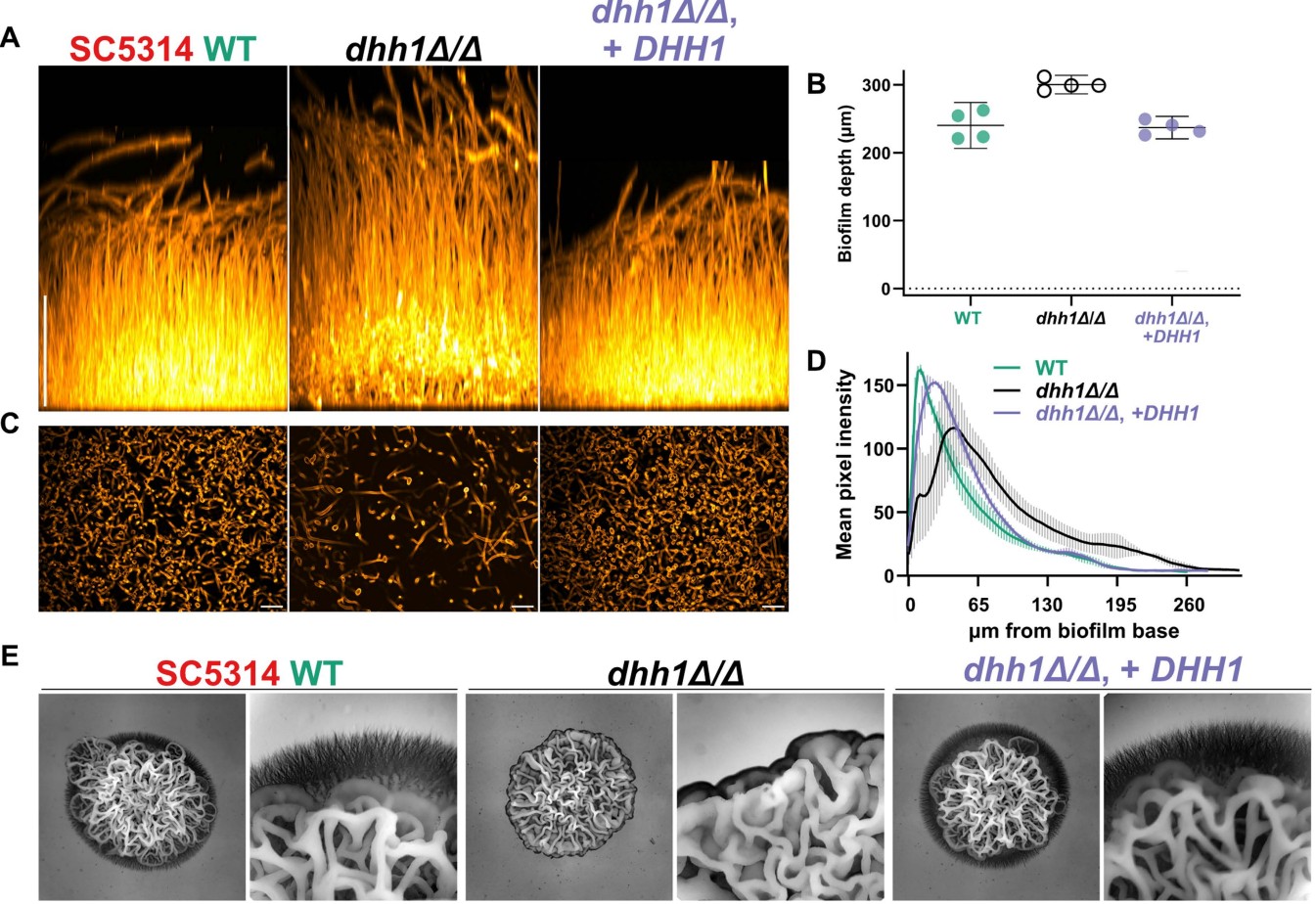

**Fig 4. Absence of *DHH1* alters SC5314 biofilm and spider colony morphology.** WT and mutant strains grown in two *in vitro* filamentation assays. A. and B. WT and mutant biofilms after a 24-hours (37°C, RPMI + 10% FBS) *dhh1Δ/Δ* has thicker biofilms. Scale bar = 100μm, 95% CI. C. Basal views 26μm from the bottom of the biofilm showed a reduced density of *dhh1Δ/Δ* cells. Scale bar = 20μm. D. Average pixel intensity (cell wall stain) from 4 replicate biofilms (SD) varied with biofilm depth. Strains with at least one copy of *DHH1* had a greater concentration of cells closer to the base of the biofilm than *dhh1Δ/Δ*. E. Colonies on spider plates require *DHH1* for hyperfilamentous margins (6 days, 37°C). 6X and 25X magnification.

## Filamentation dependent transcriptome remodeling is reduced in *dhh1Δ/Δ*

We next investigated the role of *DHH1* in gene regulation during filamentation. Comparing RNA-seq results from WT SC5314 grown in RPMI and YPD showed the known extensive transcriptome remodeling that occurs during filamentation, with 2,488 downregulated and 2,496 upregulated genes (Fig 6A and S3 Table). Hyphal cell wall proteins and ubiquitination factors such as *HWP1* and *BUL1* were upregulated in WT RPMI compared to WT YPD (Fig 6A). GO terms related to ribosomal RNA, ribosomal proteins, and translation elongation were all enriched in genes downregulated in WT hyphae, strongly indicating a general reduction in mRNA translation (Fig 6B). The majority of upregulated GO terms in WT RPMI were enriched in carbohydrate transporters responding to the shift from rich YPD to carbohydrate poor RPMI (Fig 6C). These results are consistent with known transcriptional changes that occur during WT filamentation [13,66–72].

We hypothesized that impaired mRNA decay in *dhh1Δ/Δ* would blunt the downregulation of a subset of genes during filamentation. Indeed, transcriptional remodeling of *dhh1Δ/Δ* was reduced with 1,795 downregulated and 1,919 upregulated genes, only 75% of the total

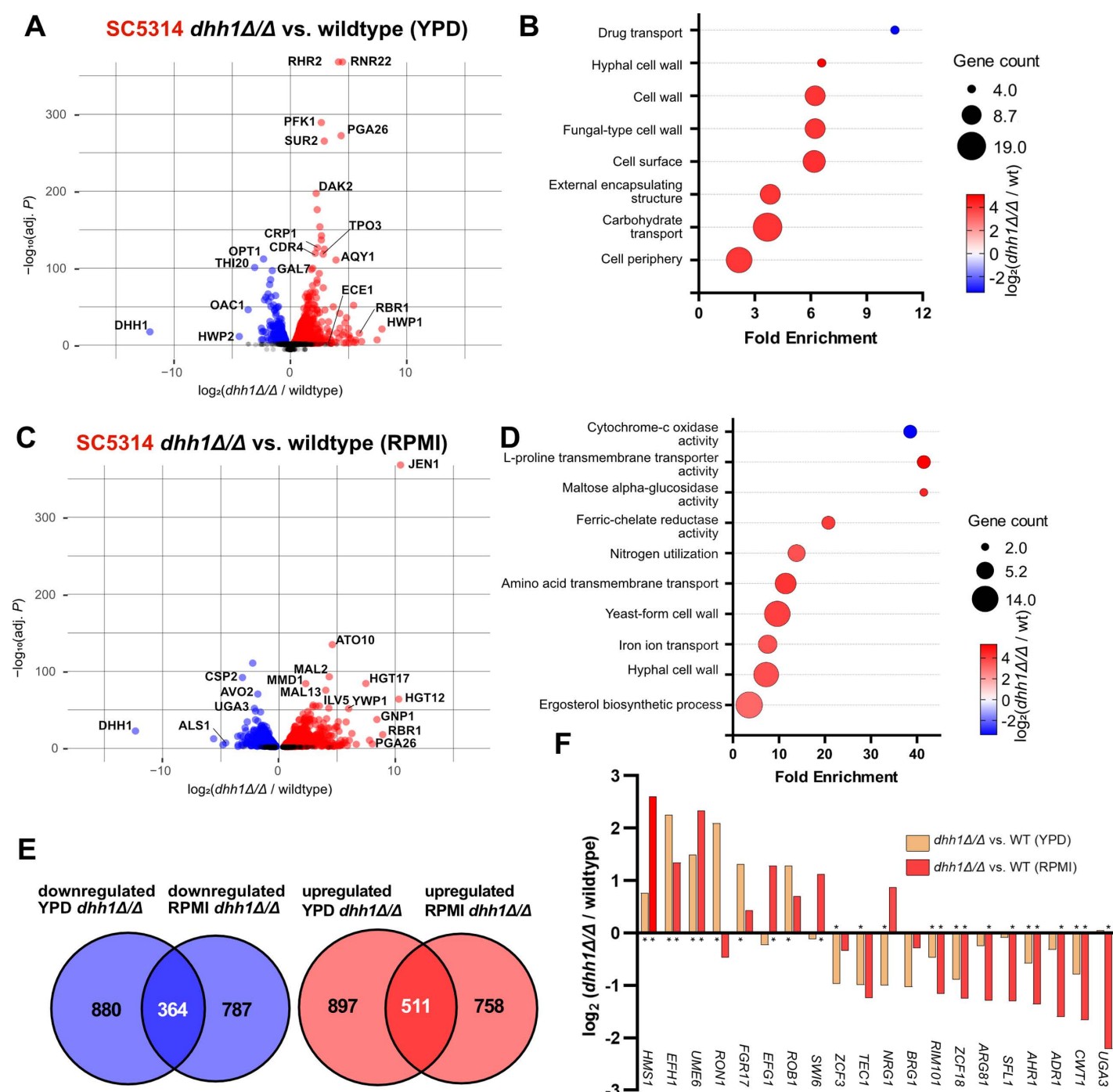

**Fig 5. RNA-seq exposed differential gene expression in SC5314 *dhh1 Δ/Δ* grown in YPD and RPMI.** A. Volcano plot depicts genes are significantly up (red) and down (blue) -regulated in *dhh1Δ/Δ* yeast-form growth compared to wildtype (Padj < 0.01). Genes associated with filamentation stress were prominently upregulated (S1 Table). B. GO terms enriched in *dhh1Δ/Δ* YPD include transport and cell wall terms (Padj < 0.1). C. Volcano plot depicts gene expression differences in *dhh1Δ/Δ* vs wildtype during early hyphal growth (S2 Table, Padj < 0.01). Genes associated with metabolite transport and the yeast/hyphal cell wall were differentially expressed in *dhh1Δ/Δ*. D. GO terms enriched in *dhh1Δ/Δ* RPMI include transport, cell wall, and metabolic terms (Padj < 0.1). E. A significant number of *dhh1Δ/Δ* transcripts were downregulated (P = 2.28e-22), and upregulated (P = 4.37e-52), respectively, in both YPD and RPMI conditions (Hypergeometric tests). However, most differentially expressed genes were environmentally specific. F. Differential expression of transcription factors regulating morphogenesis in *dhh1Δ/Δ* in YPD and RPMI (* indicates Padj < 0.01; DEseq2).

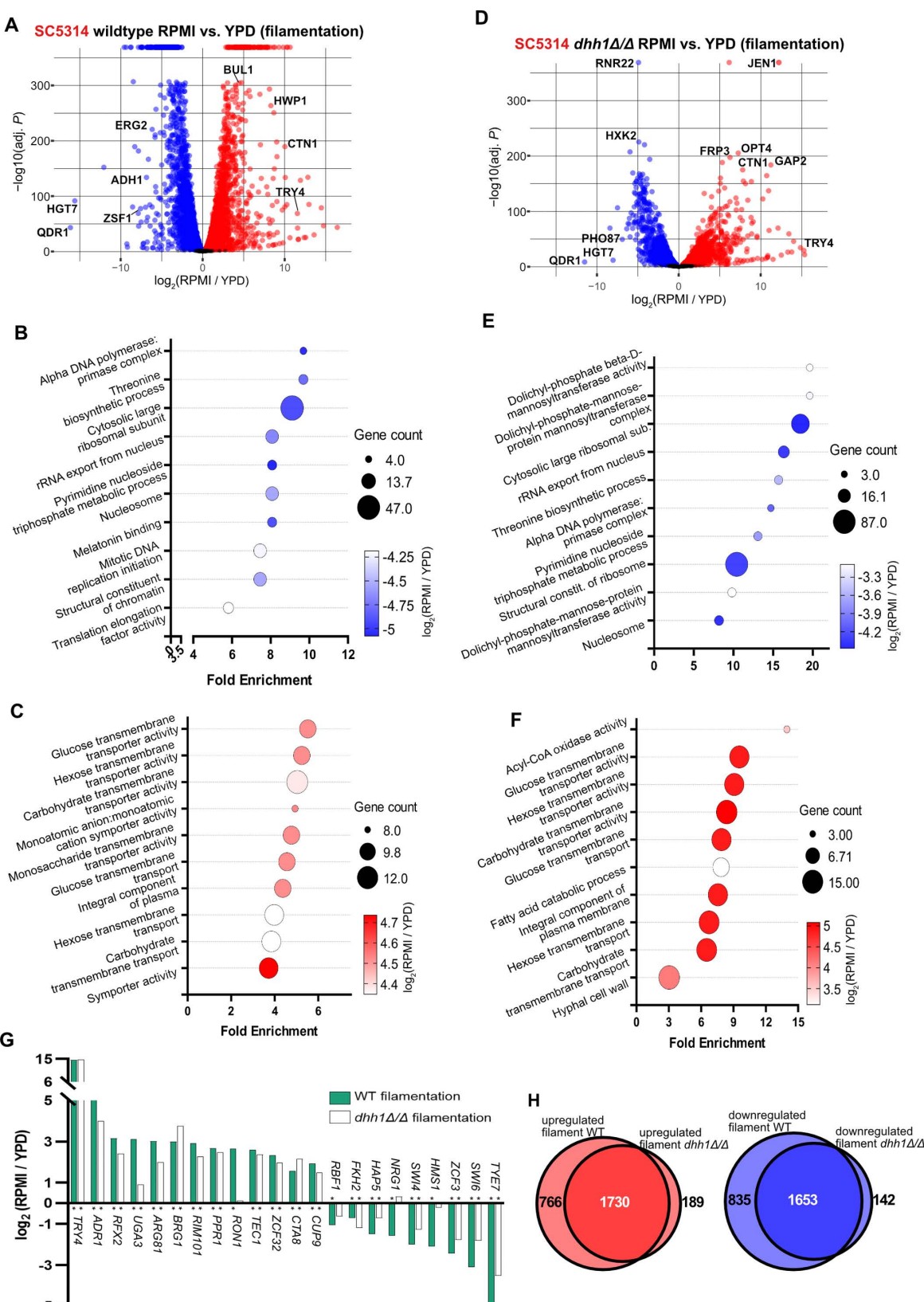

**Fig 6. RNA-seq reveals transcriptome differences between SC5314 wildtype and *dhh1 Δ/Δ* during filamentation.** A. Volcano plots comparing the transcriptomes of WT SC5314 grown in YPD and RPMI demonstrate extensive remodeling during filamentation (S3

Table, Padj < 0.01). B. Expected GO terms were downregulated, and C. upregulated during WT filamentation (Padj < 0.1). D. Comparing the transcriptomes of *dhh1Δ/Δ* grown in YPD and RPMI showed impaired remodeling during filamentation (S4 Table, Padj < 0.01). E. Similar GO terms were downregulated, and F. upregulated in filamenting *dhh1Δ/Δ* and WT (Padj < 0.1). Unlike wildtype, *dhh1Δ/Δ* upregulated terms related to lipid catabolism during filamentation. G. Transcription factors regulating morphogenesis had blunted remodeling in *dhh1Δ/Δ*. (* indicates Padj < 0.01; DEseq2). H. Most genes differentially expressed during *dhh1Δ/Δ* filamentation are subsets of the WT remodeling response. A significant number of genes were downregulated or upregulated in both filamenting *dhh1Δ/Δ* and wild type (Hypergeometric tests, P = 0).

number of genes remodeled in WT (Fig 6D). Genes uniquely differentially expressed in WT filamentation were related to metabolism and transmembrane transport (downregulated) and nucleic acid decay (upregulated). The number of up and downregulated genes during WT filamentation was almost equal with only 8 more genes upregulated compared to downregulated (Fig 6A). However, *dhh1Δ/Δ* had 124 fewer genes downregulated compared to upregulated, possibly due to impaired mRNA degradation (Fig 6D). Most of the GO terms enriched in downregulated genes in filamenting *dhh1Δ/Δ* were the same as WT (cf. Fig 6B and 6E). Similarly, most of the GO terms enriched in upregulated genes in filamenting *dhh1Δ/Δ* were also enriched in filamenting WT cells (cf. Fig 6C and 6F). Uniquely, terms related to lipid catabolism were upregulated in *dhh1Δ/Δ* but not WT (cf. Fig 6C and 6F). Changes in the expression of morphology-related transcription factors were generally less pronounced in *dhh1Δ/Δ* than WT. For some genes this was due to increased basal expression in YPD (Figs 5F and 6G). Interestingly, unlike in WT, the filamentation repressor gene *NRG1* was not significantly downregulated in filamenting *dhh1Δ/Δ*, suggesting a mechanism for continued repression of select hyphal genes (Fig 6G). The vast majority of differentially expressed genes in filamenting *dhh1Δ/Δ* were a subset of WT remodeling (Hypergeometric test, P = 0; Fig 6H). In general, SC5314 *dhh1Δ/Δ* had a more modest transcriptome remodeling response during filamentation than WT.

## Absence of *DHH1* alters growth, heat tolerance, and filamentation in diverse strains

Recent studies have identified considerable heterogeneity in regulatory networks that control filamentation in diverse *C. albicans* strains [14,73]. We next investigated the morphological impacts of *DHH1* deletion in four additional strains representing the major *C. albicans* clades [58], that we have previously used in studies of gene regulation [14]. In all backgrounds, we found *dhh1Δ/Δ* impaired YPD colony growth at both 30°C and 37°C, and only a single copy of *DHH1* was necessary for a WT appearance (Fig 7A). Interestingly, the extent of *dhh1Δ/Δ* growth reduction varied with the background strain. For example, P75010 *dhh1Δ/Δ* colonies appeared similar in size to WT by Day 3, while P76067 and P57055 *dhh1Δ/Δ* colonies were growth impaired across all temperatures and time points (Fig 7A). A similar trend was observed in liquid YPD, with P75010 exhibiting a relatively modest decrease in growth (Fig 7B). While none of the other *dhh1Δ/Δ* strains phenocopied the extensive wrinkled morphology of SC5314 *dhh1Δ/Δ* grown at 37°C on YPD plates (Fig 2D), they each diverged from WT with distinct texture in the center of the colony (Fig 7C). In the absence of *DHH1,* hyperfilamentous backgrounds (P76067, P87) lost their distinct invasive margins on spider media (Fig 7D), while less filamentous strains (P75010, P57055) produced *dhh1Δ/Δ* colonies with wrinkling reminiscent of SC5314 *dhh1Δ/Δ* (Figs 4E and 7D). Thus, as in SC5314, loss of *DHH1* in four diverse *C. albicans* strains reduced yeast growth, altered colony morphology under heat stress, and produced phenotypically non-WT colonies on spider media. These data support a central role in growth and filamentation for the RNA decay factor Dhh1 across *C. albicans* strain backgrounds.

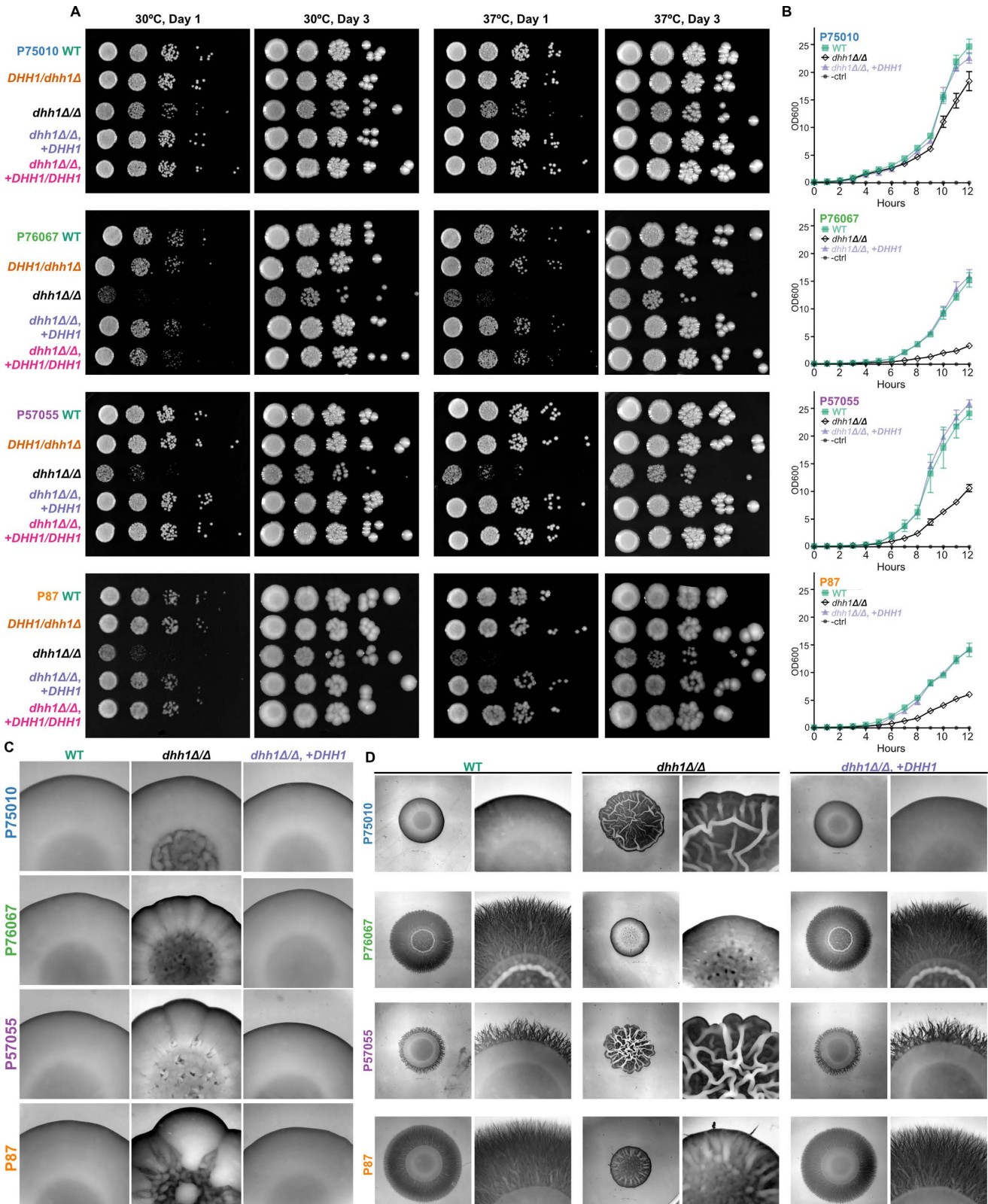

**Fig 7. *DHH1* is required for wildtype growth and colony morphology in diverse strains.** As in SC5314, DHH1 is required for wildtype growth, heat tolerance, and filamentation on spider media. Additionally, *dhh1Δ/Δ* impaired growth to a greater extent in hyperfilamentous strains. A. *dhh1Δ/Δ*

resulted in slow colony growth at 30°C and 37°C on YPD plates, the extent of which varied with background strain. B. *dhh1Δ/Δ* reduced growth in liquid YPD (30°C). WT and mutant strains were grown in triplicate. Impairment of *dhh1Δ/Δ* exponential growth varied with strain and was rescued by complementation (*dhh1Δ/Δ, +DHH1*). Error bars show CI = 95%. C. Colonies lacking *DHH1* have a non-WT textured morphology during mild heat stress (4 days, YPD agar, 37°C). 25X magnification. D. Spider colony morphology varies between strains and requires *DHH1* for WT phenotype (6 days, 37°C). 6X and 25X magnification.

### SC5314 and P57055 transcriptomes exhibit analogous responses to *DHH1* deletion

Our RNA-seq results indicated that *dhh1Δ/Δ* disrupts the expression of a wide variety of genes and blunts the induction of the hyphal gene regulatory program in SC5314. To validate these effects and identify genes whose expression is impacted by *dhh1Δ/Δ* in other genetic backgrounds, we next used RNA-seq to identify differentially expressed genes in P57055 yeast (30°C, YPD) and early hyphae (37°C, RPMI + 10% FBS, 4 hours) in triplicate (see methods for details). We chose the P57055 strain because of the four strains it was the second most phylogenetically distant from SC5314 [58] while displaying a similar WT filamentation capacity and *dhh1Δ/Δ* phenotypic aberrations (Figs 7 and S2B–F). As in SC3514, loss of *DHH1* in P57055 extensively altered the transcriptome in both yeast and hyphae (S4 Fig and S5 and S6 Tables) as well as during filamentation (S5 Fig and S7 and S8 Tables). Comparison of the RNA-seq data from SC5314 and P57055 grown in YPD, revealed a core of down (457) and upregulated (632) genes in *dhh1Δ/Δ* strains compared to their respective wildtype (Fig 8A and S12 Table). Strikingly, this core set was enriched with GO terms relating to catabolism, metabolism, and hyphal cell wall (Fig 8B-C). The effects of Dhh1 deletion on gene expression were more similar during growth in RPMI, with 883 genes down and 901 genes upregulated in both deletion strains (Fig 8D and S13 Table). Genes downregulated in both deletion strains were enriched for GO terms relating to transcription, transport, and zinc binding (Fig 8E), while metabolic terms and yeast-form cell wall were enriched in genes upregulated (Fig 8F) in both *dhh1Δ/Δ* strains grown in RPMI. Interestingly, morphology related TFs were differentially expressed in both *dhh1Δ/Δ* strains (Figs 5F and S4F). For example the hyphal repressor *NRG1* was significantly downregulated in YPD *dhh1Δ/Δ* SC5314 (Padj = 0) and P57055 (Padj = $3.96e^{-15}$) (S1 and S5 Tables). While regulators of hyphal growth *EFH1* and *UME6* were upregulated in both *dhh1Δ/Δ* strains in YPD and RPMI (Figs 5F and S4F). These results define a core set of genes and pathways dependent on Dhh1 in divergent *C. albicans* strains (S12 and S13 Tables).

Finally, we examined the impact of *dhh1Δ/Δ* on filamentation in SC5314 and P57055. In wildtype strains, transcriptome remodeling was highly correlated (2,085 genes downregulated and 2,084 genes upregulated) (Fig 8G) and largely involved known transcriptional changes that occur during filamentation (Fig 8H–I) [13,66–72]. In comparison, the transcriptional response to filamentation in the two *dhh1Δ/Δ* strains was somewhat less consistent with 1,375 genes downregulated and 1,268 genes upregulated (Fig 8J). As in filamenting SC5314 *dhh1Δ/Δ* alone (Fig 6B-C, 6E-F), the majority of the overlapping enriched GO terms in filamenting *dhh1Δ/Δ* SC5314 and P57055 (Fig 8K–L) were a subset of the WT terms (Fig 8H–I). In both filamenting *dhh1Δ/Δ* strains, the expression of morphology-related transcription factors was blunted compared to WT (Figs 6G and S5G). Notably, genes downregulated in both *dhh1Δ/Δ* deletion strains were enriched in hyphal cell wall functions (Fig 8K), suggesting a common defect in appropriate induction of hyphal wall programs during filamentation. Together, these results indicate that loss of *DHH1* causes consistent defects in the morphological transition between yeast and hyphal growth in two diverse clinical isolates of *C. albicans* (Fig 9A-B).

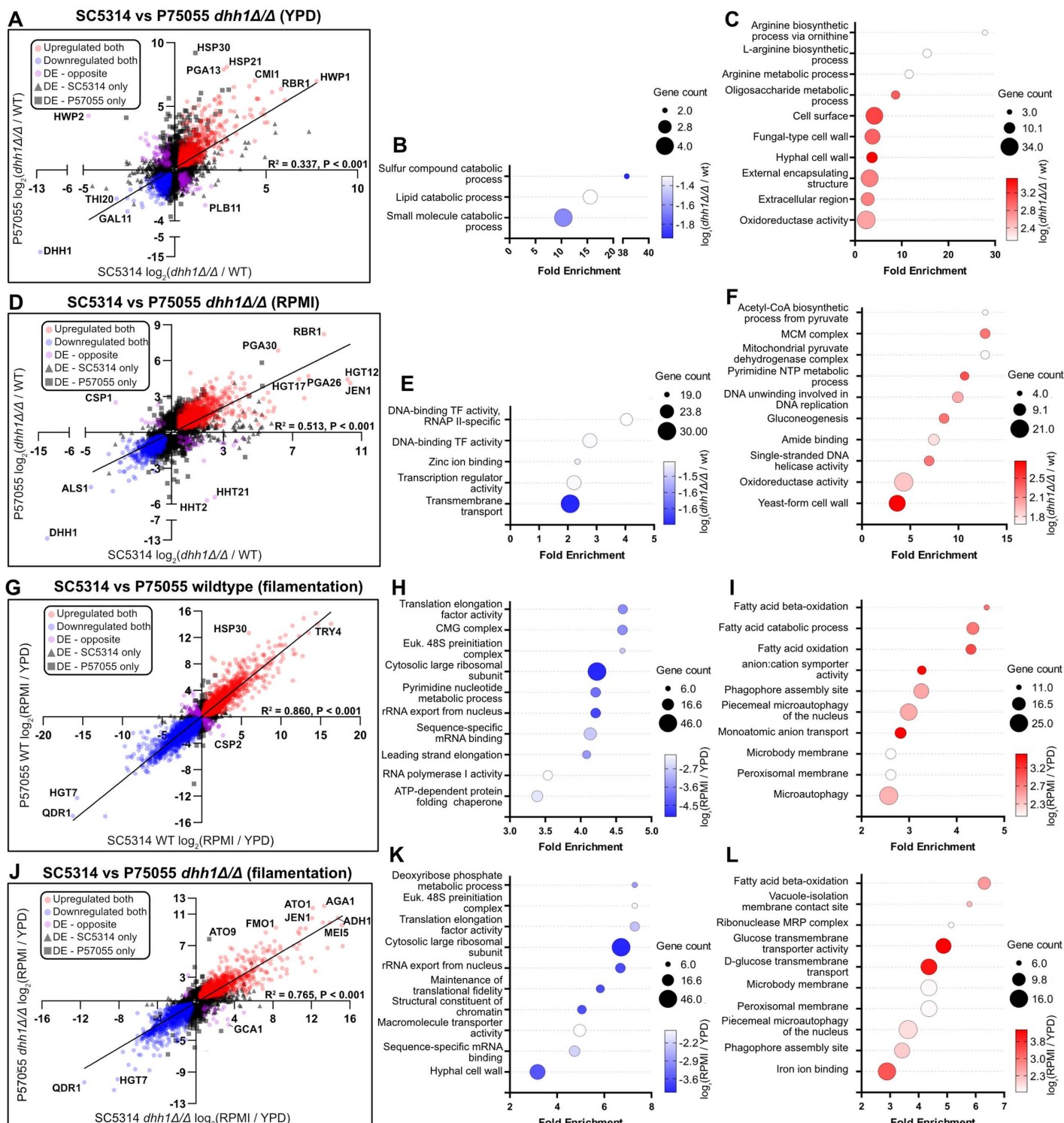

**Fig 8. Differentially expressed genes and GO terms in *dhh1 Δ/Δ* and WT strains correlated between SC5314 and P57055.** A. Scatter plot of DE genes (Padj < 0.01) in both SC5314 *dhh1Δ/Δ* and P57055 *dhh1Δ/Δ* in YPD is moderately correlated. GO analysis of core genes B. down and C. upregulated in both *dhh1Δ/Δ* strains in YPD, emphasize terms related to the cell wall and biosynthesis. D. Scatter plot of DE genes (Padj < 0.01) in both SC5314 *dhh1Δ/Δ* and P57055 *dhh1Δ/Δ* in RPMI has a greater correlation than in YPD reflecting a common stress response. GO analysis of core genes E. downregulated and F. upregulated in both *dhh1Δ/Δ* strains in RPMI. G. Scatter plot of DE genes in both WT SC5314 and WT P57055 during filamentation (Padj < 0.01) is highly correlated. GO analysis of core genes H. down and I. upregulated during SC5314 and P57055 WT filamentation shows expected regulatory changes. J. Scatter plot of DE genes in SC5314 *dhh1Δ/Δ* and P57055 *dhh1Δ/Δ* during filamentation are still correlated (Padj < 0.01). GO analysis of core genes K. down and L. upregulated are generally a subset of WT except for downregulation of hyphal wall genes.

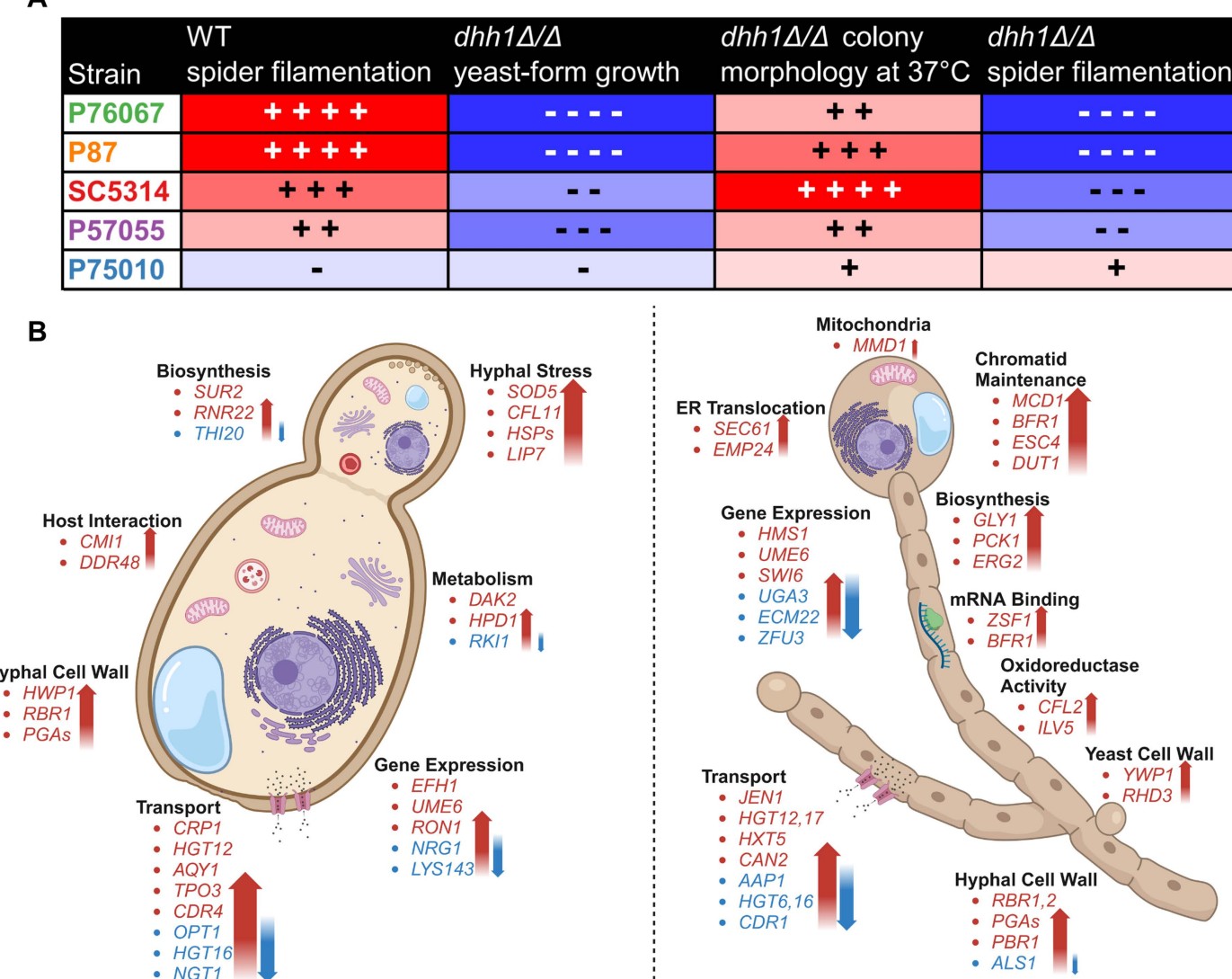

| Strain | WT spider filamentation | *dhh1Δ/Δ* yeast-form growth | *dhh1Δ/Δ* colony morphology at 37°C | *dhh1Δ/Δ* spider filamentation |
|---|---|---|---|---|
| P76067 | + + + + | - - - - | + + | - - - - |
| P87 | + + + + | - - - - | + + + | - - - - |
| SC5314 | + + + | - - | + + + + | - - - |
| P57055 | + + | - - - | + + | - - |
| P75010 | - | - | + | + |

**Fig 9. Phenotypic and transcriptomic outcomes in diverse *dhh1 Δ/Δ* strains.** Despite differences between WT background strains, loss of *DHH1* is associated with specific morphological and transcriptomic outcomes across diverse clades. A. Hyperfilamentous strains are more phenotypically sensitive to the loss of *DHH1*. B. Diagram of overlapping transcripts and terms in SC5314 *dhh1Δ/Δ* and P57055 *dhh1Δ/Δ* in YPD and RPMI. Under yeast-form growth conditions *dhh1Δ/Δ* fails to repress the hyphal stress response. Under hyphal growth conditions *dhh1Δ/Δ* presents a misregulated transcriptome including persistence of yeast features. Genes in red were significantly upregulated, while genes in blue were significantly downregulated (Padj < 0.01). Diagram produced using Biorender.

## Discussion

Foundational research on the genetic bases of filamentation largely focused on transcription factors [9] such as *BRG1* [74], *UME6* [75], and *EFG1* [12]. More recently, growing evidence for post-transcriptional regulation of filamentation has added an additional layer of control [15,16,22]. Previous studies reported that loss of a single *DHH1* copy impairs filamentation [35] and the requirement of *EDC3* for WT stress tolerance and filamentation [36,37]. These results suggested PB factors have different functions in *C. albicans* as compared to their homologs in *S. cerevisiae* and *S. pombe* [35,43,56,76,77]. However, the molecular methods originally used to generate *C. albicans DHH1/dhh1Δ* and *edc3Δ/Δ* can lead to confounding

effects that require complementation controls [78–80]. Unlike *S. cerevisiae* [46,48,81], the gene regulatory effects of *DHH1* deletion, both during yeast and hyphal growth, were not previously characterized in *C. albicans*. Here we investigated the roles of PB factors Dhh1 and Edc3 in *C. albicans* growth, filamentation, and gene expression across multiple strains.

Our work helps refine our understanding of Dhh1 and Edc3 in PB formation. Previous studies reported Edc3 was necessary for observable PBs during *C. albicans* glucose starvation [36]. Contrary to this, *S. cerevisiae* PBs condense in the absence of many individual protein factors, including Dhh1 and Edc3 [44]. Our findings align with the roles of these genes in *S. cerevisiae* and other eukaryotes. In *C. albicans*, PBs condense during heat shock without Dhh1 or Edc3 (Figs 1F, 1G, S1B and S1C). Our results suggest that, as in *S. cerevisiae* [27,44,82], the absence of multiple scaffolding proteins would be required to completely disassemble *C. albicans* P-bodies [27,44,82]. Additionally, we found Dhh1 was both present in PBs and cytoplasmically diffuse, suggesting a PB-independent function in the cytoplasm (Fig 1B), as in *S. cerevisiae* [44,45]. For example, in *S. cerevisiae*, impaired ATPase activity of Dhh1 caused it to accumulate in PBs and impaired mRNA decay of specific transcripts possibly due to decreased concentration in the cytoplasm [25,45]. Thus, *C. albicans* PBs may share many of the well documented functions and characteristics of their homologous condensates from *S. cerevisiae*.

Our phenotypic results differ from previous work. First, haploinsufficiency in filamentation was previously reported for *DHH1/dhh1Δ* in BWP17 [35]. In contrast, we found that only homozygous *dhh1Δ/Δ* mutations had aberrant filamentation in five diverse *C. albicans* strains (Figs 3,4, and 7D). Additionally, a single copy of *DHH1* complements these growth defects (Figs 3,4, and 7D). This suggests *DHH1* is not a rate limiting factor for mRNA decay. Additionally, we found that *EDC3* is not required for heat tolerance or filamentation in five different strains, contrary to results from BWP17 [36,37] (S2 Fig). While *EDC3* may not be essential for filamentation, retaining this gene likely confers an as of yet unknown evolutionary advantage to *C. albicans*. For example, Edc3 contributes to suppressing the misexpression of starvation genes in *S. cerevisiae* [54]. Although we found different phenotypes for deletion of *DHH1* and *EDC3* than in prior studies, this earlier work used a deletion approach that can cause off-target effects [78–80] and complementation studies confirming the deletions were not performed [35,36,83]. Our results highlight the power of targeted deletion through CRISPR-Cas9 and the importance of complementation analysis of genetic phenotypes.

Recently, a number of studies have reported considerable heterogeneity in the phenotypes of diverse *C. albicans* strains, which can vary depending on environmental conditions [58,84,85]. Comparison of phenotypes across diverse strains has also been used to identify core functions of genes [14,73]. We used this strategy to compare the functions of Dhh1 in filamentation in SC5314 and four additional strains. After growing diverse *dhh1Δ/Δ* strains on spider media, we found deletions in hyperfilamentous strains produced colonies that failed to invade agar, yet presented other aspects of hyphal morphology (Figs 4E and 7D). Interestingly, in strains with minimal filamentation potential (P75010 and P57055), *dhh1Δ/Δ* presented colonies with more pronounced wrinkling (Fig 7D). Furthermore, *dhh1Δ/Δ* in non-filamentous P75010 had a minimal reduction in yeast-form growth rates as compared to highly filamentous strains like P76067 (Fig 7A–B). One intriguing possibility is that highly filamentous strains are more reliant on mRNA decay to prevent the inappropriate expression of stress response and filamentation genes (Fig 9A). In conclusion, *dhh1Δ/Δ* impacts growth and colony morphology in multiple backgrounds in a manner correlated with their intrinsic filamentation propensity.

We also investigated the effects of *dhh1Δ/Δ* on the transcriptomes of *C. albicans* under both yeast and filamentous growth. Strikingly, we found the absence of *DHH1* induces major transcriptomic changes in both yeast (YPD) and hyphae (RPMI) in SC5314 and P57055 (Figs

5 and S4). These changes suggest an "identity crisis", such that the deletion strains express hyphal genes under yeast-growth conditions (Fig 9B). In YPD, *dhh1Δ/Δ* upregulates genes normally associated with hyphal virulence such as *RHR2* [86], *ECE1* [5], and *HWP1* [87] in SC5314 and P57055 (Figs 5A, S3C, S3D and S4A and S1 and S5 Tables). However, *ECE1* and *RHR2* upregulation is more modest in *dhh1Δ/Δ* P57055 and only approaches statistical significance (Padj = 0.076 and 0.013), respectively. (S1 and S5 Tables). Under both yeast and hyphal conditions, genes upregulated by *dhh1Δ/Δ* are enriched in cell wall and cell surface GO terms in both SC5314 and P57055 (Figs 5,6, S4 and S5). The resulting effect is expression of cell wall genes that are unsuitable to the growth condition (Fig 9B). For example, in both strains *dhh1Δ/Δ* upregulates yeast cell wall protein *YWP1* and downregulates hyphal adhesion *ALS1* in RPMI (Figs 5C, S3E, and S4C and S2 and S6 Tables). These results support a model in which Dhh1 prevents the misexpression of genes involved in the transition between yeast and hyphal growth.

A number of transcription factors are known to regulate filamentation [9,68,75]. Our results show that DHH1 deletion led to inappropriate expression of several key transcriptional regulators in both SC5314 and P57055 (Figs 5F and S4F). Hyphal repressor *NRG1* [62,63] was downregulated in YPD, potentially de-repressing downstream hyphal associated genes across both *dhh1Δ/Δ* strains (Fig 5F and S1 and S5 Tables). Similarly, regulators of hyphal growth *EFH1* and *UME6* were upregulated in both *dhh1Δ/Δ* strains in YPD and RPMI (Figs 5F and S4F). We find the upregulation of *UME6* particularly interesting because *UME6* constitutive expression produces hyphae under non-filament inducing conditions [75]. Furthermore, *UME6* has an extremely long 5' UTR mediating translational efficiency, and it has been proposed that sequences in the *UME6* 5' UTR could direct its transcripts to PBs [88]. The misregulation of these transcription factors could directly upregulate hyphal cell wall genes and contribute to the textured phenotype seen in all our *dhh1Δ/Δ* strains at 37°C (Figs 2D and 7C). Interestingly, under filamentation conditions SC5314 and P57055 *dhh1Δ/Δ* strains upregulated select hyphal transcription factors (e.g., *UME6*, *SWI6*) more than their respective WT strains (Figs 5F and S4F). We propose that loss of *DHH1* leads to pleiotropic hyphal morphology through alterations in the expression of hyphal transcription factors and their downstream regulatory networks (Figs 7D and S4F).

In many species stochastic transcriptional noise serves as a form of "bet-hedging" for acute environmental stressors [38–40]. For example, within a population of *C. albicans*, individual cells heterogeneously express the hyphal transcription factor Efg1, adapting colonization to host immune status [89]. Our data support a model where *C. albicans* adapts to diverse host microenvironments by transcribing multiple stress-associated genes, while simultaneously buffering expression via mRNA decay. This strategy promotes rapid responses to diverse stressors without investing energy towards environmentally inappropriate proteomic and phenotypic remodeling. Consistent with this model, we find blunted transcriptome remodeling during filamentation in both *dhh1Δ/Δ* strains as compared to their respective wildtype strains (Figs 6A, 6D, S5A, and S5D). We postulate this results from upregulation of stress-response genes in YPD and defective mRNA decay in the absence of *DHH1*. Further work is needed to test this model of decay-mediated suppression of noisy stress-response genes in *C. albicans* as a mechanism to regulate appropriate environmental adaptation.

As with our phenotype results, our transcriptome analysis of *C. albicans DHH1* deletion strains suggests a similar regulatory role as its homologs in other fungi. For example, in *C. neoformans*, the Dhh1 homologue Vad1 is required for virulence and suppresses the transcript of mating pheromone *MFα* under vegetative conditions [51,90,91]. This suggests Dhh1/Vad1-mediated RNA decay for environmentally suitable gene expression. In *S. cerevisiae*, Dhh1 has been reported to repress genes associated with nutrient limitation under optimal

growth conditions [48]. Additionally, it has been proposed that transcripts with slow translation and poor codon optimality are targeted for Dhh1 dependent mRNA decay [46,47]. Our results suggest rare codons do not mark genes for Dhh1-mediated mRNA decay in *C. albicans,* as genes upregulated in SC5314 and P57055 *dhh1Δ/Δ* had higher codon adaptation index (CAI) values than downregulated genes (S6 Fig). However, we found both *C. albicans dhh1Δ/Δ* strains upregulate genes involved in cell wall and non-preferred carbon/nitrogen metabolism – pathways similar to those that were de-repressed in *S. cerevisiae dhh1Δ* [48]. Thus, our results are consistent with the observation that Dhh1 suppresses genes associated with nutrient limitation in *S. cerevisiae.* We hypothesize that *C. albicans* Dhh1 and its homologues in other fungi support environmentally optimal gene expression by buffering noisy transcription through mRNA decay.

Altogether, our work identifies an important role for Dhh1 in regulating filamentation in *C. albicans* and implicates post-transcriptional regulation in the control of key regulators of hyphal growth. Future studies are needed to elucidate the features of transcripts repressed by Dhh1 in *C. albicans* under different growth conditions. For example, 4-thioU pulse-chase experiments could identify changes in RNA stability in the *dhh1Δ/Δ* mutant to further investigate if Dhh1 promotes decay of noisy stress-response transcripts. Additionally, an auxin inducible degradation system [92] knocking down Dhh1, could distinguish direct targets of Dhh1 mediated decay from downstream stress-responses to dysregulated RNA metabolism. Regardless of the outcomes of such studies, our work clearly defines an important role for Dhh1 in regulating filamentous growth and gene expression in *C. albicans.*

## Materials and methods

### Strains and media

All strains (S9 Table) were maintained as 15% glycerol stocks stored at -80°C. Before all experiments, strains were streaked on YPD agar (MP Biomedicals) and grown for two days at 30°C. Single colonies were picked and grown in liquid YPD (2% Bacto Peptone, 2% dextrose, 1% yeast extract) on a rotator overnight at 30°C. For PB imaging in live yeast (Fig 1A–C,F–G and S1A–C Fig), we used complete media (CM) (0.68% yeast nitrogenous base, 0.079% CSM complete AA mixture, 2% dextrose) to minimize background fluorescence. The growth media for planktonic filamentation and biofilm assays was RPMI 1640 (Gibco, minus phenol red) with 10% FBS (Gibco). For PB (Edc3-mNeonGreen) imaging during filamentation (Fig 1E), we used 100% RPMI without phenol red due to background fluorescence of FBS. We evaluated filamentous colony morphology with spider plates [93] (1% nutrient broth, 1% D-mannitol, 0.2% $K_2HPO_4$, 2% agar).

### Strain construction

WT strains and plasmids pV1093 and pRS316 were used for CRISPR-Cas9 genome editing [57, 94]. mNeonGreen [95] was appended to the C-terminus of *EDC3* using lithium acetate transformation and the SAT1 FLP/FRT cassette (pRS316) [94] without the CRISPR-Cas9 system. For all other gene insertions (including *DHH1*-mScarlet (S7A Fig)) and our whole-gene deletions, we used transient CRISPR-Cas9 [57] with the SAT1 FLP/FRT cassette [94] which leaves a 34 bp FRT scar in the genome. Optimal Cas9 guide RNAs and PAM sites were selected from a database of unique sites [96]. The SAT1 FLP/FRT cassettes with C-terminal *EDC3*-mNeonGreen and *DHH1*-mScarlet tags included sequences homologous to the 3' end of the target genes and a GGSG linker [95]. To completely delete *EDC3* and *DHH1*, our cassettes used homologous sequences upstream and downstream of the CDS (S7B Fig). Additionally, for the *DHH1* complement (*dhh1Δ/Δ* background) we inserted *DHH1* fused to the *TPI1* promoter at the *MDR1* locus replacing the entire gene (S7C Fig).

For plasmid transformations, single colonies were picked from 2 day YPD plates and incubated in 5mL YPD on a rotator overnight at 30°C. The next day, the overnight culture was diluted to $OD_{600} = 0.2$ in 50mL YPD and grown to $OD_{600} = 0.8$ (30°C, 300 RPM). Cells were collected by centrifugation (1,600 RCF, 5 minutes) in a 50mL falcon tube, washed with 5mL sterile $dH_2O$, resuspended 500μL LATE solution (0.1M LiAc/TE) and transferred to a 1.5mL microcentrifuge tube. 1μg of each PCR product (sgRNA, caCas9, SAT1 FLP/FRT) and 10μL boiled calf thymus DNA were added and cells were rotated for 30 minutes at 30°C before adding 700μL freshly made PLATE solution (800μL 50% PEG 4,000, 100μL 10X TE, 100μL 1M LiAC). Cells were then rotated overnight at 30°C. The next day, cells were gently mixed and incubated at 44°C for 15 minutes. The cells were harvested by centrifugation (1,500 RCF, 30 seconds), washed and resuspended in 2mL YPD. Cells were then incubated for 5 hours at 30°C on the rotator in 15mL falcon tubes. After this recovery period, cells were harvested by centrifugation (1,600 RCF, 3 minutes and resuspended in 200μL YPD. 100μL of resuspended cells were plated on each of two YPD plates with 400μg/mL nourseothricin (clonNAT/NTC; Goldbio #N-500-1) and incubated at 30°C for 48 hours. We extracted gDNA [57,97] and verified insertion of SAT1 FLP/FRT and homo/heterozygosity in colonies by PCR.

Transformed colonies were incubated in a 15mL falcon tube with 10mL YPM (2% Bacto Peptone, 2% maltose, 1% yeast extract) overnight to induce excision of the SAT1 FLP/FRT cassette. Overnight YPM cultures were serially diluted by $1:10^5$ to $1:10^6$ and 100μL was spread on YPD + 40μg/mL clonNAT plates and incubated at 30°C. After 48 hours, small colonies were selected, indicative of SAT1 FLP excision. All strains were validated by gDNA PCR for cassette insertion, homo/heterozygosity, and excision of SAT1 [57,97]. We used *TDH3* as the control for gDNA extraction and amplification. All strains, plasmids, guide RNA and PAM sites as well as primers for strain construction and verification are listed in S9 Table.

## Phenotype assays

For PB imaging in live cells, strains were grown overnight in a rotator at 30°C in CM, diluted to an $OD_{600}$ of 0.1, and subcultured (30°C, CM, 300 RPM) for 4 hours before loading into an Bioptechs FCS2 with a coverslip coated with 2% ConA to maintain desired temperature. All $OD_{600}$ measurements were taken with a GENESYS 20 spec with 1mL plastic cuvettes. For PB imaging during heat shock (Figs 1B,1C,1F,1G and S1), cells were incubated in CM for 10 minutes at 46°C. For imaging PBs during starvation (Fig 1D), SC5314 mNeonGreen tagged strains were incubated in PBS at room temperature for 3 hours. To image PBs (Edc3-mNeonGreen) during filamentation (Fig 1E), we used 100% RPMI minus phenol red. All other filamentation assays used RPMI supplemented with 10% FBS.

For spot plates, strains were grown overnight at 30°C in 5mL of liquid YPD on a rotator, diluted to an $OD_{600}$ of 0.5 and serially diluted 10-fold in YPD. YPD plates were spotted with 5μL of diluted cells and incubated for 1-3 days at 30°C and 37°C prior to imaging with a Bio-Rad ChemiDoc Touch Imaging System. To quantify growth in liquid YPD, strains were grown overnight at 30°C in YPD on a rotator, diluted to an $OD_{600}$ of 0.1 in 25mL YPD and incubated for 12 hours at 30°C. Strains were grown in triplicate and $OD_{600}$ was measured hourly with a GENESYS 20 visible spec with 1mL plastic cuvettes. Colony morphology under heat stress (4-days, YPD, 37°C) and filamentation conditions (6-days, spider media, 37°C) was assayed similarly prepared in the same manner. Strains were grown overnight in a rotator at 30°C in 3mL liquid YPD, harvested by centrifugation (1,000 RCF, 5 minutes), washed three times with 6mL PBS, and diluted to $OD_{600} = 0.33$ in 1mL PBS before spotting 3μL on YPD/spider plates.

To evaluate planktonic filamentation we grew strains overnight at 30°C in liquid YPD, diluted them to an $OD_{600}$ of 0.2 the following morning and subcultured them to an $OD_{600}$ of 0.6 in 25mL YPD before centrifuging (2,800 RCF, 5 minutes) the entire volume. We washed

the cell pellet with 5mL RPMI + 10% FBS and induced filamentation (25mL RPMI + 10% FBS, 37°C, 300 RPM). After 4 and 16 hours, cells were centrifuged (2,800 RCF, 5 minutes), fixed with 2mL 4% formalin, and washed three times with 2mL PBS. 500μL of the washed cells in PBS were stained the sample with 200μg/mL calcofluor white. The fixed and stained cells were allowed to adhere to a glass slide treated with polyallylamine (PA) and washed with PBS prior to imaging

To survey biofilm growth, we grew cells overnight at 30°C in 5mL YPD on a rotator and diluted them to an $OD_{600}$ of 0.33 in 1.5mL RPMI + 10% serum. In four wells 200μL of diluted cells were added to a 96-well ibidi plate coated with polyallylamine (PA) and incubated for 90-minutes at 37°C for adherence. After 90 minutes, the media was removed leaving only adherent cells and incubated with 400μL fresh RPMI + 10% serum. After a 24-hour incubation at 37°C, cells were fixed with 350μL 4% formalin, washed twice with 400μL PBS, and stained with 400μL ConA Alexa Fluor 594 conjugate (200μg/mL) overnight. The following morning we removed the ConA Alexa Fluor 594 conjugate and washed the stained biofilms with 400μL PBS prior to adding 400μL 50% TDE in PBS and incubating it for 30 minutes. The 50% TDE solution was removed and 400μL100% TDE solution was applied before confocal imaging.

Colony morphology under heat stress (4-days, YPD, 37°C) and filamentation conditions (6-days, spider media, 37°C) was assayed similarly. Strains were grown overnight in a rotator at 30°C in 3mL liquid YPD, harvested by room temperature centrifugation (1,000 RCF, 5 minutes), washed three times with 6mL PBS, and diluted to $OD_{600}$ = 0.33 in 1mL PBS before spotting 3μL on YPD/spider plates.

## Imaging and processing

Single-cell fluorescence, DIC, and transmitted light images were captured with a Zeiss Axiovert 200M microscope using a 100X Plan-NEOFLUAR, 25X Plan-Apochromat, or 40X Plan-Apochromat objective. For PB images in Fig 1, all of the strains were in a SC5314 background and imaged live. Temperature was maintained on the stage using a Bioptechs FCS2 chamber. Only Fig 1D is a confocal image captured with a 50-micron confocal slit wheel and processed with a Z-projection. The remaining images in Fig 1 consist of single planes of transmitted light and fluorescence. As such, we were unable to quantify PB number or volume per cell in those images. PB images in Fig 1 were processed in Fiji by applying the Find Edges tool in the transmitted light images for cell outlines, and background blur was reduced in the fluorescence images using Subtract Background with a rolling ball radius of 30 pixels. Transmitted and fluorescence images were overlaid using the Merge Channels tools. Replicates of the dual labeled strain (Edc3-mNG, Dhh1-mS) are in S1A Fig in addition to condensates in Dhh1-mS, *edc3Δ/Δ* (S1B Fig) and Edc3-mNG, *dhh1Δ/Δ* (S1C Fig).

Spot plate images were adjusted for brightness/contrast in Fiji for increased visibility (Figs 2A, 2B, 7A, and S2A). For the colonies under heat stress (4-days, YPD, 37°C) and filamentation conditions (6-days, spider media, 37°C), images were collected with a Wild M8 Heerbrugg Stereo Microscope at 6X and 25X and an iPhone 11 (Figs 2D, 4E, 7C, 7D, and S2F). In Fiji colony images were converted to 8-bit and the LUT was inverted to improve resolution of colony texture. To ensure comparability of colony size all 6X images were cropped to 1,500 pixels$^2$ and 25X images were cropped to 1,900 pixels$^2$ without any resizing. From DIC images in Fig 2E, cell and vacuole areas were measured in Fiji using the freehand tool in the Regions of Interest (ROI) manager. A total of 100 cells across at least 4 fields of view were quantified to generate the graphs Fig 2F-H.

For the 4 and 16 hour planktonic filamentation assays we imaged short fluorescence confocal stacks (Figs 3A, 3C, and S2B). In Fiji background blur was reduced in the fluorescence

images using Subtract Background with a rolling ball radius of 30 pixels prior to Z projection. We quantified the extent of filamentation by measuring length by width in 100 cells in at least 4 fields of view using the straight and segmented line tools in the Regions of Interest (ROI) manager (Figs 3B, 3D, and S2C).

For the biofilm growth assays, we took 25X fluorescence confocal stacks from the base to the apical region of the biofilm (Figs 4A and S2D). In Fiji we converted the images to 32 bit and subtracted the background with a rolling ball radius of 13 pixels and set any resulting negative values to zero. To generate a side view of the biofilm we resliced the rectified image, Z projected and rescaled the Y values. The depth of the biofilms was quantified using the straight line tool in the Regions of Interest (ROI) manager (Figs 4B and S2E). To estimate cell density throughout the biofilm we used pixel brightness as a proxy from the Plot Z-Axis Profile function that generates a mean brightness (of the cell wall stain) for each slice in the stack (Fig 4D).

## RNA-extraction and sequencing

All experimental strains and conditions were assayed in triplicate (SC5314 WT YPD, SC5314 WT RPMI, SC5314 *dhh1Δ/Δ* YPD, SC5314 *dhh1Δ/Δ* RPMI) and (P57055 WT YPD, P57055 WT RPMI, P57055 *dhh1Δ/Δ* YPD, P57055 *dhh1Δ/Δ* RPMI) for a total of 12 samples in each strain background. Strains were grown overnight at 30°C in liquid YPD and diluted to an $OD_{600}$ of 0.2 in liquid YPD the next morning. Sub-cultured cells were grown in 120mL YPD at 30°C with 300 RPM shaking until the $OD_{600}$ reached 0.8. 20mL of the culture was collected in a 50mL falcon tube and centrifuged at 4°C (3,220 RCF, 3 minutes). The supernatant was removed and the cells were frozen at -80°C. 80mL of yeast cells were washed with PBS and resuspended in 80mL pre-warmed RPMI + 10% serum, followed by incubation at 37°C and 300 RPM. After 4 hours, 20mL of early hyphae was collected by sterile filtration (500mL Nalgene Rapid-Flow Sterile Disposable Filter Unit with PES 0.2 μm), washed on the filter twice with 5mL ice cold RNA buffer (4M NaCl, 1MTris-HCl pH7.5, EDTA), and frozen at -80°C. Cell pellets were stored at -80°C.

RNA extraction was adapted from Cravener and Mitchell, 2020 [60]. Frozen cells in falcon tubes were resuspended in 1mL ice-cold RNA buffer (4M NaCl, 1MTris-HCl pH7.5, EDTA). 500 uL of resuspended cells were mixed with 250μL zirconia beads and 500 uL acid phenol:chloroform pH4.5 (Ambion #AM9722) in 2mL microcentrifuge tubes and subjected to 10 cycles of vortexing and chilling on ice for one minute each. After centrifugation at room temperature (16,873 RCF, 2 minutes), the aqueous phase was removed and back-extracted with an equal volume of chloroform-isoamyl alcohol 24:1 (Sigma Aldrich #C0549). The resulting aqueous phase was removed again, added to 3 volumes of ice cold 100% ethanol and precipitated at -80°C overnight. The precipitated RNA was centrifuged at 4°C (20,817 RCF, 15 minutes). RNA pellets were washed with 1,800μL 70% ice cold ethanol and resuspended in 100μL ultrapure water. Total RNA was quantified with a Thermo Scientific NanoDrop. Genomic DNA was removed from 5μg RNA using the invitrogen TURBO DNA-free kit according to manufacturer's specifications. After the DNase treatment, RNA was re-extracted by adding 250μL RNA buffer, 500μL TRizol (Invitrogen #15596026), and 250μL chloroform-isoamyl alcohol 24:1 (Sigma Aldrich #C0549). This was vortexed and centrifuged at 4°C(20,817 RCF, 15 minutes). The aqueous phase was removed and added to an equal volume of 100% ice cold ethanol and recovered using the RNA Clean & Concentrator kit from Zymo Research according to the manufacturers specifications. RNA was eluted in 60μL ultrapure water. RNA concentrations (range = 62.6ng/μL-105.5ng/μL) were measured by NanoDrop and $RIN^e$ was quantified using an Agilent 2200 TapeStation System (range = 7.8$RIN^e$ -9.9$RIN^e$). Strand-specific,150 bp-paired ends were sequenced by Novogene using the NEBNext Poly(A)

mRNA Magnetic Isolation Module (NEB #E7490) to isolate mRNA and the NEBNext Ultra II Directional RNA Library Prep Kit for Illumina (NEB #E7760S/L) and sequenced an Illumina NovaSeq X-Plus Platform.

## Transcript alignment, quantification, and quality control

Reads were aligned to the diploid *C. albicans* genome release 22 using HISAT2 version 2.2.1 [98] with the options (--known-splicesite-infile --rna-strandness R --max-intronlen 1400). SAMTools version 1.15.1 [99] used to select and sort primary hits (samtools sort), generate a bam.bai index (samtools index) and deeptools was used to generate positive (bamCoverage --normalizeUsing BPM --exactScaling -bs 1 --filterRNAstrand=forward) and negative (bamCoverage --normalizeUsing BPM --exactScaling -bs 1 --filterRNAstrand=reverse) strand specific bigWig coverage tracks for IGV version 2.8.2 [100]. For each background, we quantified the number of transcripts in each strain (WT, *dhh1Δ/Δ*) and growth condition (YPD, RPMI) with the coverageBed function of BEDTools version 2.30.0 [101]. All 12 experimental groups for SC5314 and P57055 were combined into a single file of raw hits for each strain (S10 and 11 Tables). Our aligned read depth ranged from 18.6 million (P57055 WT, RPMI) to 37.5 million (SC5314 dhh1Δ/Δ, RPMI) aligned reads with a mean aligned read depth of 26.6 million.

For quality control, coverage tracks of select genes are shown in S4 Fig. We confirmed that *DHH1* transcripts were present in WT cells and absent in *dhh1Δ/Δ* under both growth conditions with IGV (S3A Fig). Using R version 4.3.1 and the edgeR release 3.20 [102] package we converted raw hits to copies to million (S8A Fig). Poorly transcribed ORFs (<1 Counts Per Million (CPM) in <3 groups) were screened out leaving only hits with ≥1 CPM in ≥ 3 groups (S8B Fig). Using the built-in R function prcomp we performed a principal component analysis, confirming replicates are clustered by strain and growth conditions (S9A-B Fig). The filtered CPM hits were analyzed with DEseq2 release 3.20 [61] with a Padj-value cutoff of 0.01 for strict identification of differentially expressed genes.

## GO analysis

For the GO term analysis of DE genes we focused on transcripts that were differentially expressed with a $\text{Log}_2\text{FC}$> +2.5 or <-2.5 in Figs 5,6,S4, and S5. However, due to the smaller number of transcripts DE in the same direction in both SC5314 and P57055 we selected transcripts with a $\text{Log}_2\text{FC}$> +1.0 or <-1.0 for Fig 8. The analysis was performed in the *Candida* Genome Database [103] CGD Gene Ontology Term Finder (http://www.candidagenome.org/cgi-bin/GO/goTermFinder). Only terms with a corrected P-value of < 0.1 are generated with this tool. However, we selected a maximum of 10 terms with the greatest fold enrichment, an overwhelming majority of which had a Padj < 0.01 from the GO analysis.

## Software and data analysis

All images were processed with Fiji [104]. Primers and transformation cassettes were designed using sequences from the *Candida* Genome Database [103] and the *Candida albicans* Umbrella Comparative genomics project [58], NCBI bioproject ID: PRJNA193498. Codon adaptation index (CAI) was calculated using CAIcal [105] and codon usage table from the *Candida* Genome database [103]. Data analysis and statistical tests were performed using GraphPad Prism version 10.0.0 for Windows [106] as well as R version 4.3.1 [107] with ggplot2 3.5.1 [108] and bioconductor packages release 3.20 including edgeR [102] and DESeq2 [61]. We used Affinity Designer for Windows Version 2.5.5, 2024and BioRender.com to compose our figures.

## Supporting information

**S1 Fig. Replicates of PBs condensation during heat stress in SC5214.** A. PB factors (Edc3-mNG and DHH1-mS) co-localized during acute heat shock (10 minutes at 46°C in CM) confirming their identity as PBs. Scale bars = 2μm. **B.** Dhh1-mS condensed in the absence of Edc3 in response to acute heat shock (10 minutes at 46°C in CM). Scale bars = 2μm. **C.** Edc3-mNG condensed in the absence of Dhh1 response to acute heat shock (10 minutes at 46°C in CM). Scale bars = 5μm.
(TIFF)

**S2 Fig. Absence of EDC3 does not impair growth or hyphal morphology in diverse strains.** WT and edc3Δ/Δ strains were subjected to growth, filamentation, biofilm, and spider plate assays. **A.** 24-hour spot plate assay of *C. albicans* WT and *edc3Δ/Δ* shows *EDC3* is not required for WT growth in all 5 strains. **B.** *edc3Δ/Δ* strains were subjected to a planktonic 4-hour filamentation assay (37°C, RPMI + 10% serum) and did not display gross inhibition of filamentation compared to WT. Scale bars = 20μm. **C.** After 4-hours of filamentation, *edc3Δ/Δ* did not have grossly impaired germ tube development. Only *edc3Δ/Δ* P75010 was significantly less hyphal than WT (N =100 cells, ****P < 0.0001, **P < 0.01, Kruskal-Wallis tests). **D.** Side-view of WT and *edc3Δ/Δ* strains after a 24-hour biofilm assay (37°C, RPMI + 10% serum). Scale bar = 100μm. **E.** Depth of 4 replicate biofilms. 95% CI. **F.** Colonies grown on spider plates for 6 days at 37°C. 6X and 25X magnification.
(TIFF)

**S3 Fig. Normalized IGV tracks for SC5314 and P57055 WT and dhh1Δ/Δ strains.** Tracks normalized by cpm were examined to verify deletions and differentially expressed genes A. *DHH1* transcripts are present in WT backgrounds and absent from *dhh1Δ/Δ*. **B.** In YPD putative trifunctional thiamine biosynthesis enzyme *THI20* is downregulated in both *dhh1Δ/Δ* strains. **C.** Hyphal wall protein (*HWP1*) transcripts are upregulated in *dhh1Δ/Δ* compared to WT under yeast-form growth conditions. **D.** Interestingly, in YPD hyphal associated candidalysin (*ECE1*) is upregulated in SC5341 *dhh1Δ/Δ*, and to a lesser extent in P57055 *dhh1Δ/Δ*. **E.** Hyphal adhesin (*ALS1*) is downregulated in both *dhh1Δ/Δ* strains in RPMI, **F.** while GPI-anchored cell wall protein (*RBR1*) is upregulated in both *dhh1Δ/Δ* strains under these conditions.
(TIFF)

**S4 Fig. RNA-seq exposed differential gene expression in P57055 dhh1Δ/Δ grown under YPD and RPMI conditions.** A. Volcano plot depicts genes significantly up (red) and down (blue) -regulated during yeast-form growth compared to wildtype (Padj < 0.01). Genes associated with filamentation stress were prominently upregulated (S5 Table). **B.** GO terms enriched in *dhh1Δ/Δ* YPD include transport and cell wall terms (Padj < 0.1). in. **C.** Volcano plot depicts gene expression differences in *dhh1Δ/Δ* vs wildtype during early hyphal growth (S6 Table, Padj < 0.01). **D.** GO terms enriched in *dhh1Δ/Δ* RPMI are related to chromatin, replication and biosynthesis (Padj < 0.1). **E.** A significant number of transcripts were downregulated (P = 0.041), and upregulated (P = 0.0178), respectively, in *dhh1Δ/Δ* in YPD and RPMI conditions (Hypergeometric tests). However, most differentially expressed genes were environmentally specific. **F.** Differential expression of transcription factors regulating morphogenesis in *dhh1Δ/Δ* in YPD and RPMI. (* indicates Padj < 0.01; DEseq2).
(TIFF)

**S5 Fig. RNA-seq reveals transcriptome differences between WT and dhh1Δ/Δ during filamentation.** A. Volcano plots comparing the transcriptomes of WT P57055 grown in YPD and RPMI demonstrate extensive remodeling during filamentation (S7 Table, Padj < 0.01). **B.**

Expected GO terms were downregulated, and **C.** upregulated during WT filamentation (Padj < 0.1). **D.** Comparing the transcriptome of *dhh1Δ/Δ* grown in YPD and RPMI showed transcriptome remodeling during filamentation (S8 Table, Padj < 0.01). **E.** Similar GO terms were downregulated, and **F.** upregulated in filamenting *dhh1Δ/Δ* and WT (Padj < 0.1). Unlike WT, *dhh1Δ/Δ* upregulated terms related to lipid catabolism during filamentation. **G.** Transcription factors regulating morphogenesis were differentially expressed during filamentation in WT and *dhh1Δ/Δ* (* indicates Padj < 0.01; DEseq2). **H.** Genes differentially expressed during filamentation overlapped between WT and *dhh1Δ/Δ* strains. Most transcripts differentially expressed during *dhh1Δ/Δ* filamentation are subsets of the WT remodeling response. A significant number of transcripts were down- (P = 0) or upregulated (P = 0) in both filamenting *dhh1Δ/Δ* and wild type (Hypergeometric tests).
(TIFF)

**S6 Fig. Poor CAI is not associated with de-repression (upregulation) in dhh1Δ/Δ.** Upregulated genes in C. albicans dhh1Δ/Δ have higher CAI values than downregulated genes. CAIs of 100 most upregulated and downregulated genes in SC5314 *dhh1Δ/Δ* and P57055 *dhh1Δ/Δ* strains grown in **A.** YPD at 30°C and **B.** RPMI + 10% FBS at 37°C. (Mann-Whitney tests. ***P < 0., ****P < 0.0001).
(TIFF)

**S7 Fig. Transient CRISPR-cas9 cloning diagrams.** Methods for A. inserting a fluorescent tag in the genome, B. deleting the entire ORF for a gene leaving only a 34 bp FRT scar, and C. generating a DHH1 complement at the MDR1 site. Diagram produced using Biorender.
(TIFF)

**S8 Fig. Filtering out poorly transcribed ORFs in normalized RNAseq hits.** ORFs with less than one CPM hit in less than 3 experimental groups were removed prior to running DEseq2. **A.** Violin plots show raw and **B.** filtered hits for each strain and condition with ≥1 CPM in ≥ 3 experimental groups.
(TIFF)

**S9 Fig. Samples cluster by strain and growth conditions.** PCA plot displaying the filtered CPM from all 12 experimental groups clustering by condition in A. SC5314 and **B.** P57055.
(TIFF)

**S1 Table. DEseq2 results comparing SC5314 YPD *dhh1Δ/Δ* vs. SC5314 YPD WT for Fig 5A.** Exponentially growing *dhh1Δ/Δ* and WT SC5314 yeast cells in YPD at 30°C. RNA extraction performed in triplicate from three separate samples.
(XLSX)

**S2 Table. DEseq2 results comparing SC5314 RPMI *dhh1Δ/Δ* vs. SC5314 RPMI WT for Fig 5C.** dhh1Δ/Δ and WT SC5314 grown under physiologically relevant filamentation conditions (RPMI + 10% serum at 37°C) for 4 hours. RNA extraction performed in triplicate from three separate samples.
(XLSX)

**S3 Table. DEseq2 results comparing SC5314 RPMI WT vs SC5314 YPD WT for Fig 6A.** WT SC5314 yeast cells in YPD at 30°C and WT SC5314 grown under physiologically relevant filamentation conditions (RPMI + 10% serum at 37°C) for 4 hours. RNA extraction performed in triplicate from three separate samples.
(XLSX)

**S4 Table. DEseq2 results comparing SC5314 RPMI *dhh1Δ/Δ* vs SC5314 YPD *dhh1Δ/Δ* for Fig 6D.** dhh1Δ/Δ yeast cells in YPD at 30°C and dhh1Δ/Δ grown under physiologically relevant filamentation conditions (RPMI + 10% serum at 37°C) for 4 hours. RNA extraction performed in triplicate from three separate samples.
(XLSX)

**S5 Table. DEseq2 results comparing P57055 YPD *dhh1Δ/Δ* vs. P57055 YPD WT for S4A Fig.** Exponentially growing dhh1Δ/Δ and WT P57055 yeast cells in YPD at 30°C. RNA extraction performed in triplicate from three separate samples.
(XLSX)

**S6 Table. DEseq2 results comparing P57055 RPMI *dhh1Δ/Δ* vs. P57055 RPMI WT for S4C Fig.** dhh1Δ/Δ and WT P57055 grown under physiologically relevant filamentation conditions (RPMI + 10% serum at 37°C) for 4 hours. RNA extraction performed in triplicate from three separate samples.
(XLSX)

**S7 Table. DEseq2 results comparing P57055 RPMI WT vs P57055 YPD WT for S5A Fig.** WT P57055 yeast cells in YPD at 30°C and WT P57055 grown under physiologically relevant filamentation conditions (RPMI + 10% serum at 37°C) for 4 hours. RNA extraction performed in triplicate from three separate samples.
(XLSX)

**S8 Table. DEseq2 results comparing P57055 RPMI *dhh1Δ/Δ* vs P57055 YPD *dhh1Δ/Δ* for S5D Fig.** dhh1Δ/Δ yeast cells in YPD at 30°C and dhh1Δ/Δ grown under physiologically relevant filamentation conditions (RPMI + 10% serum at 37°C) for 4 hours. RNA extraction performed in triplicate from three separate samples.
(XLSX)

**S9 Table. List of strains, plasmids, PAM sites, and primers used in this study.** This table includes relevant information concerning our strain generation and validation.
(XLSX)

**S10 Table. Raw RNA-seq hits for SC5314 WT and *dhh1Δ/Δ in* YPD and RPMI.** Unfiltered raw aligned RNA-seq hits used for DEseq2 input.
(XLSX)

**S11 Table. Raw RNA-seq hits for P57055 WT and *dhh1Δ/Δ* in YPD and RPMI.** Unfiltered raw aligned RNA-seq hits used for DEseq2 input.
(XLSX)

**S12 Table. Core genes differentially expressed in the same direction for YPD SC5314 *dhh1Δ/Δ* and YPD P57057 *dhh1Δ/Δ*.** In YPD, both *dhh1Δ/Δ* strains had 457 genes down-regulated (blue circles Fig 8. A) and 632 genes upregulated (red circles Fig 8A). Data from DEseq2 results in S1 and S5 Tables.
(XLSX)

**S13 Table. Core genes differentially expressed in the same direction for RPMI SC5314 *dhh1Δ/Δ* and RPMI P57057 *dhh1Δ/Δ*.** In RPMI, both *dhh1Δ/Δ* strains had 883 genes down-regulated (blue circles Fig 8D) and 901 genes upregulated (red circles Fig 8D). Data from DEseq2 results in S2 and S6 Tables.
(XLSX)

## Acknowledgments

We would like to thank Gemma E. May for her cloning expertise and support as well as the members of the McManus laboratory for helpful discussions.

## Author contributions

**Conceptualization:** Melissa A. Tosiano, C. Joel McManus.

**Data curation:** Melissa A. Tosiano, C. Joel McManus.

**Funding acquisition:** C. Joel McManus.

**Investigation:** Melissa A. Tosiano, Frederick Lanni.

**Methodology:** Melissa A. Tosiano, Frederick Lanni, Aaron P. Mitchell.

**Resources:** Melissa A. Tosiano, Frederick Lanni, Aaron P. Mitchell.

**Supervision:** Frederick Lanni, C. Joel McManus.

**Writing – original draft:** Melissa A. Tosiano, C. Joel McManus.

**Writing – review & editing:** Melissa A. Tosiano, Aaron P. Mitchell, C. Joel McManus.

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
