## [Decision Letter · Decision Letter 0]

16 Aug 2024

Dear Dr McManus,

Thank you very much for submitting your Research Article entitled 'Roles of P-body factors in Candida albicans filamentation and stress response' to PLOS Genetics.

The manuscript was fully evaluated at the editorial level and by independent peer reviewers. The reviewers appreciated the attention to an important problem, but raised some substantial concerns about the current manuscript. Based on the reviews, we will not be able to accept this version of the manuscript, but we would be willing to review a much-revised version. We cannot, of course, promise publication at that time.

If you decide to revise the manuscript for further consideration at PLOS Genetics, please aim to resubmit within the next 60 days, unless it will take extra time to address the concerns of the reviewers, in which case we would appreciate an expected resubmission date by email to plosgenetics@plos.org.

If present, accompanying reviewer attachments are included with this email; please notify the journal office if any appear to be missing. They will also be available for download from the link below. You can use this link to log into the system when you are ready to submit a revised version, having first consulted our Submission Checklist .

PLOS has incorporated Similarity Check , powered by iThenticate, into its journal-wide submission system in order to screen submitted content for originality before publication. Each PLOS journal undertakes screening on a proportion of submitted articles. You will be contacted if needed following the screening process.

To resubmit, log into your Editorial Manager account and select the option 'Revise Submission' in the 'Submissions Needing Revision' folder.

We are sorry that we cannot be more positive about your manuscript at this stage. Please do not hesitate to contact us if you have any concerns or questions.

Yours sincerely,

Guilhem Janbon

Guest Editor

PLOS Genetics

Eva Stukenbrock

Section Editor

PLOS Genetics

Reviewer's Responses to Questions

**Comments to the Authors:**

Reviewer #1: The work by Tosiano MA et al. investigates the role of P-body component, Dhh1, the DEAD-box helicase involved in mRNA decapping, in filamentous growth of C. albicans. The Authors initially also investigate another P-body component, Edc3, but find no effect of deleting EDC3 on filamentation and focus most of the study on Dhh1. Filamentation is key to virulence of C. albicans. Thus, the subject of this study is significant. The Authors find that cells lacking Dhh1 have aberrant morphology and impaired filamentation. Previous studies implicated Dhh1 and Edc3 in filamentation in C. albicans. This study confirms the involvement of Dhh1 but suggests that Edc3 may not be critical for filamentation. In addition, this study investigates the transcriptome of the cells lacking Dhh1 and finds interesting patterns of gene expression consistent with previous studies in S. cerevisiae which also reveal a hyphae-inducing program in the mutant in non-filamenting conditions. Overall, this is a well-executed and well-described study that thoroughly evaluates transcriptional changes resulting from the deletion of DHH1 gene. It reinforces the role of this gene in mRNA level regulation and the connection to filamentation. This study forms a solid base for future studies that will address roles in gene regulation and filamentation of other specific genes revealed here.

Specific comments/suggestions:

1. Ln 12-14: Are those studies in C. albicans? this should be specified here.

2. Ln 89: perhaps the Authors could explain why specifically these two factors are of interest?

3. Ln 106-107: this is just one genetic background – perhaps in other genetic backgrounds/strains the Edc3 may have a more substantial role, in contrast to SC5314.

4. Figure 1 D and E - as there is no quantification included, it is unclear if the localizations are unchanged in the absence of the gene or perhaps diminished as compared to the wt

5. Figure 1. DIC/nemarski or an outline of the cell (for confocal) would be helpful.

6. Figure 1 legend: “10 m at” ; “3 hours of”

7. Is it possible that the dhh1 homozygous deletion mutant is an aneuploid or tetraploid?

8. Ln 225: 75% still remodeled just like in the wt - what is unique/common about those remaining 25% that failed to be remodeled?

9. Ln 247: the Authors state: "while the overlap of genes upregulated in both conditions was not significant" then later in ln 249 "commonly upregulated genes were involved in... (but this overlap was not significant? – seems contradictory?)

10. Fig 8 legend - it should be "yeast-form"

Reviewer #2: In this manuscript, Tosiano et al. investigated the role of the P-body factor DHH1 in Candida albicans and its impact on filamentous morphology and the transcriptome during filamentation. The authors demonstrate that Dhh1 and Edc3 localized to P bodies, and that P bodies can form independently of Dhh1 or Edc3 in Candida albicans, consistent with the functions of these genes in Saccharomyces cerevisiae. The study also provided evidence that DHH1 is essential for normal growth and WT filamentation behaviors, but that haploinsufficiency does not yield deficiencies in these assays, whereas deletion of EDC3 does not affect growth or heat sensitivity, in contrast to earlier studies on these genes. Complementation of the null phenotypes by the DHH1 gene suggests the present work is correct, and that previous results may be artifacts attributable to the strategy used for deletions in earlier work. Interestingly, deletion of DHH1 resulted in pseudohyphal morphology without the stressors normally required for filamentation. However, in the presence of the triggers for hyphal formation, deletion of DHH1 led to differences in germ tube formation, invasion of media, and biofilm structure. Analysis of the transcriptome of cells lacking DHH1 in yeast-type media showed differences in gene expression that were similar to WT cells undergoing filamentation, suggesting derepression of expression of some genes in the mutant without stress. In the presence of stress triggers that lead to filamentation, the DHH1 null had additional differences in gene expression. Overall, the work suggests a role for Dhh1 in remodeling the transcriptome to regulate filamentation of C. albicans.

The paper is important and will be of interest to researchers in the RNA/translational control and fungal pathogenesis fields. In general the claims are very well-supported by the data, but there are some minor issues and suggestions that do not require new experiments. In most of the figures, the number of replicates, number of fields analyzed and number of cells for microscopy, significance determinants aren’t explicitly indicated. Since the work is in contrast to prior published work, it is critical to show documentation of replicates and significance where used so the strength of claims are clear to the field.

Figure 1-Number of replicates, number of cells imaged need to be indicated, and for 1D and 1E, showing more cells would strengthen the claim that Dhh1 and Edc3 are not independently required for PB condensation. If the authors wanted to make the claim that the factors enhance P-body levels, they would need to quantify granules per cell, but they don't make the claim.

To support the claim that dhh1∆/∆ cells were twice the size of the DHH1-containing strains, the authors should provide quantifications of cell size from a substantial number of imaged cells, either mean length and width and/or plots of distributions as shown in a latter figure, or mean area -/+ error, so significance of the difference can be evaluated.

Please indicate the determinant of significance for coloring dots on the volcano plots in Figure 6/7.

Please indicate what was used for GO term determination in figure 7.

For figure 6B/DF, please indicate the significance cutoffs used for generating the gene lists for GO and assigning number of gene up/downregulated.

Please indicate in Figures 6F and 7G whether changes in the expression of each transcript in the bar graphs were statistically significant.

Line 114-nomenclature error

Line 273-the data being discussed don’t necessarily indicate RNA decay independent of P bodies, just that there is a P-body-independent function of Dhh1.

Line 280, nomenclature error

Line 282, the authors may wish to remove the contraction for formal writing conventions.

Line 284-extra e

Line 286-the data don’t report on RNA levels for Edc3, so can’t comment on whether Edc3 enhances decapping in this organism, just that it is associated with P bodies. Authors could make the comment more speculative.

Reviewer #3: The manuscript investigates how Candida albicans morphology and stress responses depend on 2 RNA-interacting proteins, Dhh1 and Edc3. These have been investigated extensively in S. cerevisiae, both for their role as P-body markers and in RNA decay. Previous data on the roles in C. albicans was scarce. The major contribution is to show that, in the laboratory strain SC5314, deletion of DHH1 alters filamentation and growth phenotypes and alters the RNA abundance of filamentation-related genes. This is shown several ways and includes complementation of phenotypes dhh1∆/∆ + DHH1. This observation seems surprising, and as stated contributes to an increasing and important body of "evidence for post-transcriptional regulation of filamentation in C. albicans".

The manuscript is generally reasonably written, and engages with most of the relevant literature including directly comparing their results to published C. albicans datasets. I think that it could be improved by shorter and more focused introduction, RNA-seq results, and discussion, to highlight the main ideas more clearly

The manuscript is lacking in addressing the generalizability of observations to other strains, in validation of the RNA-seq data, multiple problems with the microscopy figure, in some aspects of data presentation, and in inadequate methods descriptions. These limitations need to be acknowledged before acceptance for publication. There are opportunities for new experiments and analyses addressing some of these limitations, that should result in a more impactful manuscript.

Generalizability to other strains. The core observation of DHH1 affecting filamentation of the SC5314 strain appears to be robust, as it's shown by liquid media assays, biofilm assays, and RNA-seq, all in 3-4 replicates. However, there is no evidence as to whether these phenotypes extend to other genetic backgrounds, which is important as extensive recent work from multiple labs argues for the importance of phenotypic variation between fungal strains. It's now known that SC5314 is atypical within C. albicans species complex. One of the authors, Aaron Mitchell, has made the importance of genetic variation a centerpiece of his conference presentations in recent years. The Mitchell lab has published several papers demonstrating that multiple transcriptional regulators in C. albicans have different effects in different genetic backgrounds. These papers include RNA-seq experiments quantifying the importance of genetic background on the results of gene deletion. This body of well-evidenced work argues that the completing the present study to the current standards of the field would involve mutating DHH1 in multiple genetic backgrounds and repeating key phenotyping assays, and ideally include RNA-seq (or otherwise RT-qPCR of a few key targets). If these experiments are not feasible within the authors' resources, they must highlight the major limitation of using only SC5314.

The discussion acknowledges a related limitation, regarding condition-dependent filamentation: "These apparently diametrical results are not uncommon in other C. albicans genetic backgrounds where the same strain may have varied filamentation phenotypes under different conditions." I agree that extending the work to other filamentation conditions, in addition to other strains, would increase the impact of the work.

Validation of the RNA-seq data, at least, testing of key biological hypotheses arising from the data. The current manuscript ends with the RNA-seq analysis, whose main results are that loss of DHH1: 1. disrupts pathways involved in metabolism and morphology, including leading to upregulation of filamentation pathways in otherwise non-filamenting conditions, and 2. reduces filamentation dependent transcriptome remodeling. This first result is striking. The manuscript presents these results at length and discusses around 2 hypotheses: "the loss of DHH1 not only impacts the direct targets of mRNA decay, but downstream regulation of the yeast-to-hyphae gene regulatory network". Is there any way that direct vs indirect effects could be tested? Any hyphal regulator that could be over- or under-expressed to suppress the effect of dhh1∆? Any direct targets of DHH1 that could be manipulated to test if this interaction is causative? Any S. cerevisiae DHH1 datasets (CLIP-seq from Mitchell et al 2013; RIP-seq from Miller et al 2018) that could be analysed to directly compare targets? As it is, it feels like the manuscript stops and speculates just after the most interesting observation, and any steps the authors could take to move the story forward here would increase the impact of the observation.

Microscopy data in figure 1 has multiple problems, that could be easily addressed. It shows only stress conditions, acute heat shock (46°C 10 min) or nutrient starvation ("3 hours of acute glucose starvation"). Non-stress conditions need to be included so that readers can judge if there is a difference. The glucose starvation condition was not clearly described in methods. Panels 1D and 1E the conditions aren't clear. There's no quantitative analysis of P-bodies/foci, and it would be very nice to know how many there are per cell or if either Edc3 or Dhh1 affects the number of foci of the other. Also, this figure is disconnected from the rest of the manuscript: what about during the YPD and filamentation conditions used in later figures? Does Dhh1 forming P-bodies matter for its effect on morphology and gene expression?

Data presentation is most of an issue for the RNA-seq where the description is patchy and not always in a helpful place for the reader. The discussion of the DGE analysis and GO enrichment is thorough, but the information needed for an RNA-seq expert to assess or reproduce the experiment is inadequate. Could the number of replicates and basic QC be mentioned in the results in addition to the bigger description in methods? How were RNA-seq libraries prepared - poly(A) enrichment or rRNA depletion, which kit? What was the depth of sequencing? Can the analysis pipeline be described more clearly, the methods aren't totally clear if edgeR or DESeq2 were used or both? Which alignment settings for HISAT2, which genome release, was the haploid or diploid genome used? Can all counts per gene in tabular format be added to NCBI GEO as well as the raw counts on SRA? The descriptions of growth conditions and sample preparation from the author link at SRA are also incomplete.

The data presentation for figures 3-5 (morphology) was overall ok. Figure 3A should add replicate or error bar information. In figure 3 It would be nice to have quantification of vacuole size given that's a conclusion. The quantifications in figure 4-5 were effective.

The methods overall need extensive additions and clarification, e.g. are these whole-ORF deletions, what linkers were used for fluorescent tagging, what were the PAM sites, numbers of replicates/labware/volumes/method of OD recording for growth curves, details or reference for acid phenol RNA extraction, parts numbers for kits and enzymes, how were images processed beyond "with Fiji", and many others.

Given the need for major clarifications and revisions, I did not go through line-by-line to suggest minor revisions.

To end: the observation that Dhh1 affects filamentation is interesting, and the wider point of the importance of post-transcriptional regulation in fungal morphology and virulence is an important direction for the field. There's good work here and some solid observations upon which to build a stronger revised manuscript.

**Have all data underlying the figures and results presented in the manuscript been provided?**

Reviewer #1: Yes

Reviewer #2: Yes

Reviewer #3: **No: ** Counts per gene for RNA-seq (but, I had trouble accessing the supplemental data files due to traveling while writing my review, so did not check underlying data for all other figures)

PLOS authors have the option to publish the peer review history of their article (what does this mean? ). If published, this will include your full peer review and any attached files.

**Do you want your identity to be public for this peer review?** For information about this choice, including consent withdrawal, please see our Privacy Policy .

Reviewer #1: No

Reviewer #2: No

Reviewer #3: No

---

## [Decision Letter · Decision Letter 1]

3 Feb 2025

PGENETICS-D-24-00773R1

Roles of P-body factors in Candida albicans filamentation and stress response

PLOS Genetics

Dear Dr. McManus,

Thank you for submitting your manuscript to PLOS Genetics. After careful consideration, we feel that it has merit but does not fully meet PLOS Genetics's publication criteria as it currently stands. Therefore, we invite you to submit a revised version of the manuscript that addresses the points raised during the review process.

Please submit your revised manuscript within 30 days Mar 05 2025 11:59PM. If you will need more time than this to complete your revisions, please reply to this message or contact the journal office at plosgenetics@plos.org. Please include the following items when submitting your revised manuscript:

We look forward to receiving your revised manuscript.

Kind regards,

Guilhem Janbon

Guest Editor

PLOS Genetics

Eva Stukenbrock

Section Editor

PLOS Genetics

Aimée Dudley

Editor-in-Chief

PLOS Genetics

Anne Goriely

Editor-in-Chief

PLOS Genetics

**Additional Editor Comments :**

Please address the minor issues raised by reviewer 3

**Journal Requirements:**

1) Thank you for stating that "Raw RNA-seq data have been deposited at the NCBI SRA under bioproject number PRJNA1133750. Processed expression data are available at NCBI GEO under accession GSE284543". Please note that, though access restrictions are acceptable now, your entire minimal dataset will need to be made freely accessible if your manuscript is accepted for publication. This policy applies to all data except where public deposition would breach compliance with the protocol approved by your research ethics board. If you are unable to adhere to our open data policy, please kindly revise your statement to explain your reasoning and we will seek the editor's input on an exemption.

2) Please upload the figures in a correct numerical order in the online submission form.

**Reviewers' comments:**

Reviewer's Responses to Questions

Reviewer #1: The Authors have addressed all of my comments adequately.

Reviewer #2: The new experiments showing effects of DHH1 on filamentous growth/colony morphology, temperature tolerance, and RNAseq (for one isolate) of clinical isolates strengthen the claim that DHH1 is critical for WT responses to clinically relevant stressors and increases the impact of the work. It is interesting that the clinical isolates are more sensitive to DHH1 deletion for growth rates and temperature sensitivity, suggesting a critical role for DHH1 in the hyperfilamentous growth phenotype of these isolates.

The authors have addressed all of my comments in the revision and I would recommend publication of the revised manuscript.

Reviewer #3: The authors have thoroughly responded to reviewer comments, adding new data, quantifying existing data, and strengthening the manuscript with effective rewriting. It is really a pleasure to see reviewers giving consistent and constructive feedback, and authors responding to this so strongly.

Addition of the extra strains substantially strengthens the manuscript, showing that phenotypes are consistent across strains with underlying variation. Figure 7 is very nice indeed and presents the data effectively, as does Figure S2 and Figure 9A. The argument in the discussion is very nice, that dhh1-deletion changes suggest an “identity crisis”. Still, could the manuscript more clearly discuss why they chose these other 4 strains out of all strains? Especially why they did the additional RNA-seq in P57055 only?

The methods descriptions are hugely improved, adding all relevant details of RNA-seq experimental design and data analysis, and data sharing including counts per gene tables. The updated descriptions of the image analysis, strain construction, and strain/plasmid/oligo table are thorough.

The manuscript should be accepted.

Some minor comments.

Line 138, "After 3 hours of acute glucose starvation in PBS" - this could be further clarified, because 3hrs in PBS is starvation for everything else as well as glucose.

Line 460, formatting problems in "EDC3mNeonGreen and DHH1mScarlet tags"

Line 536, "SC5313 background"

Supplemental information spreadsheet Tables S1-S13 - it would be great to add a sheet zero with the table of contents for this in the same file, to aid navigation.

**Have all data underlying the figures and results presented in the manuscript been provided?**

Reviewer #1: Yes

Reviewer #2: Yes

Reviewer #3: Yes

PLOS authors have the option to publish the peer review history of their article (what does this mean? ). If published, this will include your full peer review and any attached files.

**Do you want your identity to be public for this peer review?** For information about this choice, including consent withdrawal, please see our Privacy Policy .

Reviewer #1: No

Reviewer #2: No

Reviewer #3: **Yes: ** Edward Wallace

**Figure resubmission:**
---

## [Editor Report · Decision Letter 2]

20 Feb 2025

Dear Dr McManus,

We are pleased to inform you that your manuscript entitled "Roles of P-body factors in Candida albicans filamentation and stress response" has been editorially accepted for publication in PLOS Genetics. Congratulations!

Yours sincerely,

Guilhem Janbon

Guest Editor

PLOS Genetics

Eva Stukenbrock

Section Editor

PLOS Genetics

Aimée Dudley

Editor-in-Chief

PLOS Genetics

Anne Goriely

Editor-in-Chief

PLOS Genetics

Comments from the reviewers (if applicable):

**Data Deposition**

http://datadryad.org/submit?journalID=pgenetics&manu=PGENETICS-D-24-00773R2

**Press Queries**

---

## [Editor Report · Acceptance letter]

PGENETICS-D-24-00773R2

Roles of P-body factors in Candida albicans filamentation and stress response

Dear Dr McManus,

We are pleased to inform you that your manuscript entitled "Roles of P-body factors in Candida albicans filamentation and stress response" has been formally accepted for publication in PLOS Genetics! Your manuscript is now with our production department and you will be notified of the publication date in due course.

With kind regards,

Anita Estes

PLOS Genetics

On behalf of:
